# Near-Optimal Sample Complexity Bounds
# for Constrained MDPs

**Sharan Vaswani**[*]
Simon Fraser University
vaswani.sharan@gmail.com

**Lin F. Yang**[*]
University of California, Los Angeles
linyang@ee.ucla.edu

**Csaba Szepesvári**
Amii, University of Alberta, DeepMind
szepesva@ualberta.ca

## Abstract

In contrast to the advances in characterizing the sample complexity for solving Markov decision processes (MDPs), the optimal statistical complexity for solving constrained MDPs (CMDPs) remains unknown. We resolve this question by providing *minimax* upper and lower bounds on the sample complexity for learning near-optimal policies in a discounted CMDP with access to a generative model (simulator). In particular, we design a model-based algorithm that addresses two settings: (i) *relaxed feasibility*, where small constraint violations are allowed, and (ii) *strict feasibility*, where the output policy is required to satisfy the constraint. For (i), we prove that our algorithm returns an $\varepsilon$-optimal policy with probability $1 - \delta$, by making $\tilde{O}\left(\frac{SA \log(1/\delta)}{(1-\gamma)^3 \varepsilon^2}\right)$ queries to the generative model, thus matching the sample-complexity for unconstrained MDPs. For (ii), we show that the algorithm's sample complexity is upper-bounded by $\tilde{O}\left(\frac{SA \log(1/\delta)}{(1-\gamma)^5 \varepsilon^2 \zeta^2}\right)$ where $\zeta$ is the problem-dependent Slater constant that characterizes the size of the feasible region. Finally, we prove a matching lower-bound for the strict feasibility setting, thus obtaining the first near minimax optimal bounds for discounted CMDPs. Our results show that learning CMDPs is as easy as MDPs when small constraint violations are allowed, but inherently more difficult when we demand zero constraint violation.

## 1 Introduction

Common reinforcement learning (RL) algorithms focus on optimizing an unconstrained objective, and have found applications in games such as Atari [23] or Go [28], robot manipulation tasks [29, 37] or clinical trials [26]. However, many applications require the planning agent to satisfy constraints – for example, in wireless sensor networks [10] where there is a constraint on average power consumption. More generally, in the constrained Markov decision processes (CMDP) framework, the goal is to find a policy that maximizes the value associated with a reward function subject to the policy achieving a return (for a second reward function) that exceeds an apriori determined threshold [3]. There has been substantial work addressing the planning problem to find a near-optimal policy in a known CMDP [8, 7, 30, 24, 1, 35]. However, since the CMDP is unknown in most practical applications, we consider the problem of finding a near-optimal policy in this more challenging setting.

There have been multiple recent approaches to obtain a near-optimal policy in CMDPs in the regret-minimization or PAC-RL settings [13, 38, 9, 19, 31, 22, 36, 12, 15, 16, 11]. These works

---

[*]Equal contribution

tackle the exploration, estimation and planning problems simultaneously. On the other hand, recent works [16, 33, 6] consider an easier, but even more fundamental problem of obtaining a near-optimal policy with access to a simulator or *generative model* [20, 18, 2]. In particular, these works assume that the transition probabilities in the underlying CMDP are unknown, but the planner has access to a sampling oracle (the generative model) that returns a sample of the next state when given any state-action pair as input. This is the problem setting we consider and *aim to obtain matching upper and lower bounds on the sample complexity of planning in CMDPs with access to a generative model*.

Given a target error $\varepsilon > 0$, the approximate CMDP objective is to return a policy that achieves a cumulative reward within an $\varepsilon$ additive error of the optimal policy in the CMDP. Previous work can be classified into two categories based on how it tackles the constraint – for the easier problem that we term *relaxed feasibility*, the policy returned by an algorithm is allowed to violate the constraint by at most $\varepsilon$. On the other hand, for the more difficult *strict feasibility* problem, the returned policy is required to strictly satisfy the constraint and achieve zero constraint violation. Except for the recent works of Wei et al. [33] and Bai et al. [6], most provably efficient approaches including those in the regret-minimization and PAC-RL settings consider the relaxed feasibility setting. For this problem, the best model-based algorithm requires $\tilde{O}\left(\frac{S^2 A}{(1-\gamma)^3 \varepsilon^2}\right)$ samples to return an $\varepsilon$-optimal policy in an infinite-horizon $\gamma$-discounted CMDP with $S$ states and $A$ actions [16], while the best model-free approach requires $\tilde{O}\left(\frac{SA}{(1-\gamma)^5 \varepsilon^2}\right)$ samples for achieving the objective [12]. On the other hand, the best known upper bounds for a model-free algorithm in the strict feasibility setting are achieved by Bai et al. [6]. In particular, their algorithm requires $\tilde{O}\left(\frac{SA}{(1-\gamma)^2 \varepsilon^2}\right)$ samples [6, Theorem 2] to output an $\varepsilon$-optimal policy. However, their analysis considers normalized reward and constraint value functions [6, Eq. 1] that lie in the $[0, 1]$ range (compared to the standard $[0, 1/1 - \gamma]$ range). This difference in the scale of the values prevents a direct comparison of their results to our sample complexity bounds. Subsequently, we show that when appropriately normalized, our sample complexity bounds are better by a $(1/1-\gamma)$ factor in both the relaxed feasibility (Section 4) and strict feasibility settings (Section 5).

Importantly, there are no lower bounds characterizing the difficulty of either the relaxed or strict feasibility problems (except in degenerate cases where the constraint is always satisfied and the CMDP problem reduces to an unconstrained MDP). To get an indication of what the optimal bounds might be, it is instructive to compare these results to the unconstrained MDP setting. For unconstrained MDPs with access to a generative model, both model-based [2, 21] and model-free approaches [27] can return an $\varepsilon$-optimal policy within near-optimal $\tilde{\Theta}\left(\frac{SA}{(1-\gamma)^3 \varepsilon^2}\right)$ sample-complexity [4]. Hence, compared to the sample-complexity for unconstrained MDPs, the best-known upper-bounds for CMDPs are worse for both the relaxed and strict feasibility settings. However, it is unclear whether solving CMDPs is inherently more difficult than unconstrained MDPs. We resolve these questions for both the relaxed and strict feasibility settings, and make the following contributions.

**Generic model-based algorithm**: In Section 3, we provide a generic model-based primal-dual algorithm (Algorithm 1) that can be used to achieve both the relaxed and strict feasibility objectives (with appropriate parameter settings). The proposed algorithm requires solving a sequence of unconstrained empirical MDPs using any black-box MDP planner.

**Upper-bound on sample complexity under relaxed feasibility**: In Section 4, we prove that with a specific set of parameters, Algorithm 1 uses no more than $\tilde{O}\left(\frac{SA}{(1-\gamma)^3 \varepsilon^2}\right)$ samples to achieve the relaxed feasibility objective. This improves upon the bounds of HasanzadeZonuzy et al. [16] and matches the lower-bound in the easier unconstrained MDP setting, implying that our bounds are near-optimal. Our result indicates that under relaxed feasibility solving CMDPs is as easy as solving unconstrained MDPs. To the best of our knowledge, these are the first such bounds.

**Upper-bound on sample-complexity under strict feasibility**: In Section 5, we prove that with a specific set of parameters, Algorithm 1 uses no more than $\tilde{O}\left(\frac{SA}{(1-\gamma)^5 \zeta^2 \varepsilon^2}\right)$ to achieve the strict feasibility objective. Here $\zeta \in (0, 1/1-\gamma]$ is the problem-dependent *Slater constant* that characterizes the size of the feasible region and influences the difficulty of the problem. Unlike Bai et al. [6], our bounds do not depend on additional (potentially large) problem-dependent quantities.

**Lower-bound on sample-complexity under strict feasibility**: In Section 7, we prove a matching problem-dependent $\Omega\left(\frac{SA}{(1-\gamma)^5 \zeta^2 \varepsilon^2}\right)$ lower bound on the sample-complexity in the strict feasibility

setting. Our results thus demonstrate that the proposed model-based algorithm is near minimax optimal. Furthermore, our bounds indicate that under strict feasibility (i) solving CMDPs is inherently more difficult than solving unconstrained MDPs, and (ii) the problem hardness (in terms of the sample-complexity) increases as $\zeta$ (and hence the size of the feasible region) decreases. To the best of our knowledge, these are first results characterizing the difficulty of solving CMDPs with access to a generative model and demonstrate a separation between the relaxed and strict feasibility settings.

**Overview of techniques**: For proving the upper bounds, we use a specific primal-dual algorithm that reduces the CMDP planning problem to solving multiple unconstrained MDPs. Specifically, by using a strong-duality argument, we show that we can obtain an optimal CMDP policy by averaging the optimal policies of a specific sequence of MDPs. For each MDP in this sequence, we use the model-based techniques from Agarwal et al. [2], Li et al. [21] to prove concentration results for data-dependent policies. This allows us to prove concentration for the optimal data-dependent policy in the CMDP, and subsequently bound the sample complexity for both the relaxed and strict feasibility problems. For the lower bound, we modify the MDP hard instances [5, 34] to handle a constraint reward. This makes the resulting gadgets significantly more complex than those required for MDPs, but we show that similar likelihood arguments can be used to prove the lower-bound.

## 2 Problem Formulation

We consider an infinite-horizon discounted constrained Markov decision process (CMDP) [3] denoted by $M$, and defined by the tuple $\langle \mathcal{S}, \mathcal{A}, \mathcal{P}, r, c, b, \rho, \gamma \rangle$ where $\mathcal{S}$ is the set of states, $\mathcal{A}$ is the action set, $\mathcal{P} : \mathcal{S} \times \mathcal{A} \to \Delta_{\mathcal{S}}$ is the transition probability function, $\rho \in \Delta_{\mathcal{S}}$ is the initial distribution of states and $\gamma \in [0, 1)$ is the discount factor. The primary reward to be maximized is denoted by $r : \mathcal{S} \times \mathcal{A} \to [0, 1]$, whereas the constraint reward is denoted by $c : \mathcal{S} \times \mathcal{A} \to [0, 1]$[2]. If $\Delta_{\mathcal{A}}$ denotes the simplex over the action space, the expected discounted return or *reward value function* of a stationary, stochastic policy[3] $\pi : \mathcal{S} \to \Delta_{\mathcal{A}}$ is defined as $V_r^\pi(\rho) = \mathbb{E}_{s_0, a_0, \dots} \left[ \sum_{t=0}^\infty \gamma^t r(s_t, a_t) \right]$, where $s_0 \sim \rho, a_t \sim \pi(\cdot|s_t)$, and $s_{t+1} \sim \mathcal{P}(\cdot|s_t, a_t)$. For each state-action pair $(s, a)$ and policy $\pi$, the reward action-value function is defined as $Q_r^\pi : \mathcal{S} \times \mathcal{A} \to \mathbb{R}$, and satisfies the relation: $V_r^\pi(s) = \langle \pi(\cdot|s), Q_r^\pi(s, \cdot) \rangle$, where $V_r^\pi(s)$ is the reward value function when the starting state is equal to $s$. Analogously, the *constraint value function* and constraint action-value function of policy $\pi$ is denoted by $V_c^\pi(\rho)$ and $Q_c^\pi$ respectively. The CMDP objective is to return a policy that maximizes $V_r^\pi(\rho)$, while ensuring that $V_c^\pi(\rho) \geq b$. Formally,

$$\max_\pi V_r^\pi(\rho) \quad \text{s.t.} \quad V_c^\pi(\rho) \geq b. \tag{1}$$

The optimal stochastic policy for the above CMDP is denoted by $\pi^*$ and the corresponding reward value function is denoted by $V_r^*(\rho)$. We also define $\zeta := \max_\pi V_c^\pi(\rho) - b$ as the problem-dependent quantity referred to as the Slater constant [12, 6]. The Slater constant is a measure of the size of the feasible region and determines the difficulty of solving Eq. (1).

For simplicity of exposition, we assume that the rewards $r$ and constraint rewards $c$ are known, but the transition matrix $\mathcal{P}$ is unknown and needs to be estimated. We note that assuming the knowledge of the rewards does not affect the leading terms of the sample complexity since learning these is an easier problem compared to the transition matrix [5, 27]. We assume access to a *generative model* or simulator that allows the agent to obtain samples from the $\mathcal{P}(\cdot|s, a)$ distribution for any $(s, a)$. Assuming access to such a generative model, our aim is to characterize the sample complexity required to return a near-optimal policy $\hat{\pi}$ in $M$. Given a target error $\varepsilon > 0$, we can characterize the performance of policy $\hat{\pi}$ in two ways:

**Relaxed feasibility**: We require $\hat{\pi}$ to achieve an approximately optimal reward value, while allowing it to have a small constraint violation in $M$[4]. Formally, we require $\hat{\pi}$ s.t.

$$V_r^{\hat{\pi}}(\rho) \geq V_r^*(\rho) - \varepsilon, \text{ and } V_c^{\hat{\pi}}(\rho) \geq b - \varepsilon. \tag{2}$$

---

[2]These ranges for $r$ and $c$ are chosen for simplicity. Our results can be easily extended to handle other ranges.
[3]The performance of an optimal policy in a CMDP can always be achieved by a stationary, stochastic policy [3]. On the other hand, for an MDP, it suffices to only consider stationary, deterministic policies [25].
[4]In general, the desired gap in the reward value can be different from the level of constraint violation.

---

**Algorithm 1:** Model-based algorithm for CMDPs with generative model

---

1 **Input**: $\mathcal{S}$ (state space), $\mathcal{A}$ (action space), $r$ (rewards), $c$ (constraint rewards), $\zeta$ (Slater constant), $N$ (number of samples), $b'$ (constraint RHS), $\omega$ (perturbation magnitude), $U$ (projection upper bound), $\varepsilon_1$ (epsilon-net resolution), $T$ (number of iterations), $\lambda_0 = 0$ (initialization).

2 For each state-action $(s, a)$ pair, collect $N$ samples from $\mathcal{P}(.|s, a)$ and form $\hat{\mathcal{P}}$.

3 Perturb the rewards to form vector $r_p(s, a) := r(s, a) + \xi(s, a)$ where $\xi(s, a) \sim \mathcal{U}[0, \omega]$.

4 Form the empirical CMDP $\hat{M} = \langle \mathcal{S}, \mathcal{A}, \hat{\mathcal{P}}, r_p, c, b', \rho, \gamma \rangle$.

5 Form the epsilon-net $\Lambda = \{0, \varepsilon_1, 2\varepsilon_1, \dots, U\}$.

6 **for** $t \leftarrow 0$ **to** $T - 1$ **do**

7      Update the policy by solving an unconstrained MDP: $\hat{\pi}_t = \arg \max \hat{V}^{\pi}_{r_p + \lambda_t c}$.

8      Update the dual-variables: $\lambda_{t+1} = \mathcal{R}_\Lambda \left[ \mathbb{P}_{[0,U]} \left[ \lambda_t - \eta \left( \hat{V}^{\hat{\pi}_t}_c(\rho) - b' \right) \right] \right]$.

9 **end**

10 **Output**: Mixture policy $\bar{\pi}_T = \frac{1}{T} \sum_{t=0}^{T-1} \hat{\pi}_t$.

---

**Strict feasibility** We require $\hat{\pi}$ to achieve an approximately optimal reward value, while simultaneously demanding zero constraint violation in $M$. Formally, we require $\hat{\pi}$ s.t.

$$V^{\hat{\pi}}_r(\rho) \geq V^*_r(\rho) - \varepsilon, \text{ and } V^{\hat{\pi}}_c(\rho) \geq b \tag{3}$$

Next, we describe a general model-based algorithm to handle both these cases, and subsequently instantiate the algorithm for the relaxed feasibility (Section 4) and strict feasibility (Section 5) settings.

## 3 Methodology

We will use a model-based approach [2, 21, 16] for achieving the objectives in Eq. (2) and Eq. (3). In particular, for each $(s, a)$ pair, we collect $N$ independent samples from $\mathcal{P}(\cdot|s, a)$ and form an empirical transition matrix $\hat{\mathcal{P}}$ such that $\hat{\mathcal{P}}(s'|s, a) = \frac{N(s'|s,a)}{N}$, where $N(s'|s, a)$ is the number of samples that have transitions from $(s, a)$ to $s'$. These estimated transition probabilities are used to form an empirical CMDP. Due to a technical requirement, (which we will clarify in the Section 6), we require adding a small random perturbation to the rewards in the empirical CMDP[5]. In particular, for each $s \in \mathcal{S}$ and $a \in \mathcal{A}$, we define the perturbed rewards $r_p(s, a) := r(s, a) + \xi(s, a)$ where $\xi(s, a) \sim \mathcal{U}[0, \omega]$ are i.i.d. uniform random variables. Finally, compared to Eq. (1), we will require solving the empirical CMDP with a constraint right-hand side equal to $b'$. Note that setting $b' < b$ corresponds to loosening the constraint, while $b' > b$ corresponds to tightening the constraint. This completes the specification of the empirical CMDP $\hat{M}$ that is defined by the tuple $\langle \mathcal{S}, \mathcal{A}, \hat{\mathcal{P}}, r_p, c, b', \rho, \gamma \rangle$. For $\hat{M}$, the corresponding reward value function (and constraint value function) for policy $\pi$ is denoted as $\hat{V}^{\pi}_{r_p}(\rho)$ (and $\hat{V}^{\pi}_c(\rho)$ respectively). In order to fully instantiate $\hat{M}$, we require setting the values of $\omega$ (the magnitude of the perturbation) and $b'$ (the constraint right-hand side). This depends on the specific setting (relaxed vs strict feasibility) and we do this in Sections 4 and 5 respectively. We compute the optimal policy for the empirical CMDP $\hat{M}$ as follows:

$$\hat{\pi}^* \in \arg \max \hat{V}^{\pi}_{r_p}(\rho) \text{ s.t. } \hat{V}^{\pi}_c(\rho) \geq b' \tag{4}$$

In contrast to Agarwal et al. [2], Li et al. [21] that consider model-based approaches for unconstrained MDPs and can solve the resulting empirical MDP using any black-box approach, we will require solving Eq. (4) using a specific primal-dual approach that we outline next. Using this algorithm enables us to prove optimal sample complexity bounds under both relaxed and strict feasibility.

First, observe that Eq. (4) can be written as an equivalent saddle-point problem – $\max_\pi \min_{\lambda \geq 0} \left[ \hat{V}^{\pi}_{r_p}(\rho) + \lambda \left( \hat{V}^{\pi}_c(\rho) - b' \right) \right]$, where $\lambda \in \mathbb{R}$ corresponds to the Lagrange multiplier for the constraint. The solution to this saddle-point problem is $(\hat{\pi}^*, \lambda^*)$ where $\hat{\pi}^*$ is the optimal empirical policy and $\lambda^*$ is the optimal Lagrange multiplier. We solve the above saddle-point problem

---

[5]Similar to MDPs [21], we can instead perturb the $Q$ function while planning in the empirical CMDP.

iteratively, by alternatively updating the policy (primal variable) and the Lagrange multiplier (dual variable). If $T$ is the total number of iterations of the primal-dual algorithm, we define $\hat{\pi}_t$ and $\lambda_t$ to be the primal and dual iterates for $t \in [T] := \{1, \ldots, T\}$. The primal update at iteration $t$ is given as:

$$\hat{\pi}_t = \arg\max \left[ \hat{V}_{r_p}^\pi(\rho) + \lambda_t \hat{V}_c^\pi(\rho) \right] = \arg\max \hat{V}_{r_p + \lambda_t c}^\pi. \tag{5}$$

Hence, iteration $t$ of the algorithm requires solving an unconstrained MDP with a reward equal to $r_p + \lambda_t c$. This can be done using any black-box MDP solver such as policy iteration. The algorithm updates the Lagrange multipliers using a gradient descent step and requires projecting and rounding the resulting dual variables. In particular, the dual variables are first projected onto the $[0, U]$ interval, where $U$ is chosen to be an upper-bound on $|\lambda^*|$. After the projection, the resulting iterates are rounded to the closest element in the set $\Lambda = \{0, \varepsilon_1, 2\varepsilon_1, \ldots, U\}$, a one-dimensional epsilon-net (with resolution $\varepsilon_1$) over the dual variables. In Section 6, we will see that constructing such an $\varepsilon_1$-net will enable us to prove concentration results for all $\lambda \in \Lambda$. The dual update at iteration $t$ is given as:

$$\lambda_{t+1} = \mathcal{R}_\Lambda \left[ \mathbb{P}_{[0,U]} \left[ \lambda_t - \eta \left( \hat{V}_c^{\hat{\pi}_t}(\rho) - b' \right) \right] \right], \tag{6}$$

where $\mathbb{P}_{[0,U]}[\lambda] = \arg\min_{p \in [0,U]} |\lambda - p|$ projects $\lambda$ onto the $[0, U]$ interval and $\mathcal{R}_\Lambda[\lambda] = \arg\min_{p \in \Lambda} |\lambda - p|$ rounds $\lambda$ to the closest element in $\Lambda$. Since $\Lambda$ is an epsilon-net, for all $\lambda \in [0, U]$, $|\lambda - \mathcal{R}_\Lambda[\lambda]| \leq \varepsilon_1$. Finally, $\eta$ in Eq. (6) corresponds to the step-size for the gradient descent update. The above primal-dual updates are similar to the dual-descent algorithm proposed in Paternain et al. [24]. The pseudo-code summarizing the entire model-based algorithm is given in Algorithm 1. We note that although Algorithm 1 requires the knowledge of $\zeta$, this is not essential and we can instead use an estimate of $\zeta$. In Appendix F, we show that we can estimate $\zeta$ to within a factor of 2 using $\tilde{O}\left( \frac{|\mathcal{S}||\mathcal{A}|}{(1-\gamma)^3 \zeta^2} \right)$ additional queries. Next, we show that the primal-dual updates in Algorithm 1 can be used to solve the empirical CMDP $\hat{M}$. Specifically, we prove the following theorem (proof in Appendix A) that bounds the average optimality gap (in the reward value function) and constraint violation for the mixture policy returned by Algorithm 1.

**Theorem 1** (Guarantees for the primal-dual algorithm)**.** For a target error $\varepsilon_{\text{opt}} > 0$ and the primal-dual updates in Eq. (5)-Eq. (6) with $U > |\lambda^*|$, $T = \frac{4U^2}{\varepsilon_{\text{opt}}^2 (1-\gamma)^2} \left[ 1 + \frac{1}{(U-\lambda^*)^2} \right]$, $\eta = \frac{U(1-\gamma)}{\sqrt{T}}$ and $\varepsilon_1 = \frac{\varepsilon_{\text{opt}}^2 (1-\gamma)^2 (U-\lambda^*)}{6U}$, the mixture policy $\bar{\pi}_T := \frac{1}{T} \sum_{t=0}^{T-1} \hat{\pi}_t$ satisfies,

$$\hat{V}_{r_p}^{\bar{\pi}_T}(\rho) \geq \hat{V}_{r_p}^{\hat{\pi}^*}(\rho) - \varepsilon_{\text{opt}} \quad ; \quad \hat{V}_c^{\bar{\pi}_T}(\rho) \geq b' - \varepsilon_{\text{opt}}.$$

Hence, with $T = O(1/\varepsilon_{\text{opt}}^2)$ and $\varepsilon_1 = O(\varepsilon_{\text{opt}}^2)$, the algorithm outputs a policy $\bar{\pi}_T$ that achieves a reward $\varepsilon_{\text{opt}}$ close to that of the optimal empirical policy $\hat{\pi}^*$, while violating the constraint by at most $\varepsilon_{\text{opt}}$. Hence, with sufficient number of iterations $T$ and by choosing a sufficiently small resolution $\varepsilon_1$ for the epsilon-net, we can use the above primal-dual algorithm to approximately solve the problem in Eq. (4). In order to completely instantiate the primal-dual algorithm, we require setting $U > |\lambda^*|$. We will subsequently do this for the the relaxed and strict feasibility settings in Sections 4 and 5 respectively. We note that in contrast to Paternain et al. [24, Theorem 3] that bounds the Lagrangian, Theorem 1 provides explicit bounds on both the reward suboptimality and constraint violation.

We conclude this section by making some observations about the primal-dual algorithm – while the subsequent bounds for both settings heavily depend on using the "best-response" primal update in Eq. (5), the algorithm does not require using the specific form of the dual updates in Eq. (6). Indeed, when used in conjunction with the projection and rounding operations in Eq. (6), we can use any method to update the dual variables (not necessarily gradient descent) provided that it results in an $O\left(T^a + \varepsilon_1 T^b\right)$ (for $a < 1$) bound on the dual regret (see the proof of Theorem 1 for the definition). Next, we specify the values of $N, b', \omega, \varepsilon_1, T, U$ in Algorithm 1 to achieve the objective in Eq. (2).

## 4 Upper-bound under Relaxed Feasibility

In order to achieve the objective in Eq. (2) for a target error $\varepsilon > 0$, we require setting $N = \tilde{O}\left( \frac{\log(1/\delta)}{(1-\gamma)^3 \varepsilon^2} \right)$, $b' = b - \frac{3\varepsilon}{8}$ and $\omega = \frac{\varepsilon(1-\gamma)}{8}$. This completely specifies the empirical CMDP $\hat{M}$

and the problem in Eq. (4). In order to specify the primal-dual algorithm, we set $U = O\left(1/\varepsilon\,(1-\gamma)\right)$, $\varepsilon_1 = O\left(\varepsilon^2(1-\gamma)^2\right)$ and $T = O\left(1/(1-\gamma)^4\varepsilon^4\right)$. With these choices, we prove the following theorem in Appendix B and provide a proof sketch below.

> **Theorem 2.** For a fixed $\varepsilon \in (0, 1/1-\gamma]$ and $\delta \in (0,1)$, Algorithm 1 with $N = \tilde{O}\left(\frac{\log(1/\delta)}{(1-\gamma)^3\varepsilon^2}\right)$ samples, $b' = b - \frac{3\varepsilon}{8}$, $\omega = \frac{\varepsilon(1-\gamma)}{8}$, $U = O\left(1/\varepsilon\,(1-\gamma)\right)$, $\varepsilon_1 = O\left(\varepsilon^2(1-\gamma)^2\right)$ and $T = O\left(1/(1-\gamma)^4\varepsilon^4\right)$, returns policy $\bar{\pi}_T$ that satisfies the objective in Eq. (2) with probability at least $1 - 4\delta$.

*Proof Sketch:* We prove the result for a general primal-dual error $\varepsilon_{\mathrm{opt}} < \varepsilon$ and $b' = b - \frac{\varepsilon - \varepsilon_{\mathrm{opt}}}{2}$, and subsequently specify $\varepsilon_{\mathrm{opt}}$ and hence $b'$. In Lemma 11 (proved in Appendix B), we show that if the constraint value functions are sufficiently concentrated (the empirical value function is close to the ground truth value function) for both the optimal policy $\pi^*$ in $M$ and the mixture policy $\bar{\pi}_T$ returned by Algorithm 1, i.e., if

$$\left|V_c^{\bar{\pi}_T}(\rho) - \hat{V}_c^{\bar{\pi}_T}(\rho)\right| \leq \frac{\varepsilon - \varepsilon_{\mathrm{opt}}}{2} \quad ; \quad \left|V_c^{\pi^*}(\rho) - \hat{V}_c^{\pi^*}(\rho)\right| \leq \frac{\varepsilon - \varepsilon_{\mathrm{opt}}}{2}, \tag{7}$$

then (i) policy $\bar{\pi}_T$ violates the constraint in $M$ by at most $\varepsilon$, i.e., $V_c^{\bar{\pi}_T}(\rho) \geq b - \varepsilon$, and (ii) its suboptimality in $M$ (compared to $\pi^*$) can be decomposed as:

$$V_r^{\pi^*}(\rho) - V_r^{\bar{\pi}_T}(\rho) \leq \frac{2\omega}{1-\gamma} + \varepsilon_{\mathrm{opt}} + \left|V_{r_p}^{\pi^*}(\rho) - \hat{V}_{r_p}^{\pi^*}(\rho)\right| + \left|\hat{V}_{r_p}^{\bar{\pi}_T}(\rho) - V_{r_p}^{\bar{\pi}_T}(\rho)\right|. \tag{8}$$

In order to instantiate the primal-dual algorithm, we require a concentration result for policy $\pi_c^*$ that maximizes the the constraint value function, i.e. if $\pi_c^* := \arg\max V_c^{\pi}(\rho)$, then we require $\left|V_c^{\pi_c^*}(\rho) - \hat{V}_c^{\pi_c^*}(\rho)\right| \leq \varepsilon + \varepsilon_{\mathrm{opt}}$. In Case 1 of Lemma 9 (proved in Appendix A), we show that if this concentration result holds, then we can upper-bound the optimal dual variable $|\lambda^*|$ by $\frac{2(1+\omega)}{(\varepsilon+\varepsilon_{\mathrm{opt}})(1-\gamma)}$. With these results in hand, we can instantiate all the algorithm parameters except $N$ (the number of samples required for each state-action pair). In particular, we set $\varepsilon_{\mathrm{opt}} = \frac{\varepsilon}{4}$ and hence $b' = b - \frac{3\varepsilon}{8}$, and $\omega = \frac{\varepsilon(1-\gamma)}{8} < 1$. Setting $U = \frac{32}{5\varepsilon\,(1-\gamma)}$ ensures that the $U > |\lambda^*|$ condition required by Theorem 1 holds. To guarantee that the primal-dual algorithm outputs an $\frac{\varepsilon}{4}$-approximate policy, we use Theorem 1 to set $T = O\left(\frac{1}{(1-\gamma)^4\varepsilon^4}\right)$ iterations and $\varepsilon_1 = O\left(\varepsilon^2(1-\gamma)^2\right)$. Eq. (8) can then be simplified as,

$$V_r^{\pi^*}(\rho) - V_r^{\bar{\pi}_T}(\rho) \leq \frac{\varepsilon}{2} + \left|V_{r_p}^{\pi^*}(\rho) - \hat{V}_{r_p}^{\pi^*}(\rho)\right| + \left|\hat{V}_{r_p}^{\bar{\pi}_T}(\rho) - V_{r_p}^{\bar{\pi}_T}(\rho)\right|.$$

Putting everything together, in order to guarantee an $\varepsilon$-reward suboptimality for $\bar{\pi}_T$, we require that:

$$\left|V_c^{\pi_c^*}(\rho) - \hat{V}_c^{\pi_c^*}(\rho)\right| \leq \frac{5\varepsilon}{4} \; ; \; \left|V_c^{\bar{\pi}_T}(\rho) - \hat{V}_c^{\bar{\pi}_T}(\rho)\right| \leq \frac{3\varepsilon}{8} \; ; \; \left|V_c^{\pi^*}(\rho) - \hat{V}_c^{\pi^*}(\rho)\right| \leq \frac{3\varepsilon}{8}$$
$$\left|V_{r_p}^{\pi^*}(\rho) - \hat{V}_{r_p}^{\pi^*}(\rho)\right| \leq \frac{\varepsilon}{4} \; ; \; \left|\hat{V}_{r_p}^{\bar{\pi}_T}(\rho) - V_{r_p}^{\bar{\pi}_T}(\rho)\right| \leq \frac{\varepsilon}{4}. \tag{9}$$

We control such concentration terms for both the constraint and reward value functions in Section 6, and bound the terms in Eq. (9). In particular, we prove that for a fixed $\varepsilon \in (0, 1/1-\gamma]$, using $N \geq \tilde{O}\left(\frac{\log(1/\delta)}{(1-\gamma)^3\,\varepsilon^2}\right)$ samples enssures that the statements in Eq. (9) hold with probability $1 - 4\delta$. This guarantees that $V_r^{\pi^*}(\rho) - V_r^{\bar{\pi}_T}(\rho) \leq \varepsilon$ and $V_c^{\bar{\pi}_T}(\rho) \geq b - \varepsilon$. $\qquad\square$

Hence, the total sample-complexity of achieving the objective in Eq. (2) is $\tilde{O}\left(\frac{SA\log(1/\delta)}{(1-\gamma)^3\varepsilon^2}\right)$. This result improves over the $\tilde{O}\left(\frac{S^2A\log(1/\delta)}{(1-\gamma)^3\varepsilon^2}\right)$ result in HasanzadeZonuzy et al. [16]. Furthermore, our result matches the lower-bound in the easier unconstrained setting [4], implying that our bounds are near-optimal. We conclude that under relaxed feasibility and with access to a generative model, solving constrained MDPs is as easy as solving MDPs. Algorithmically, we do not require constructing an optimistic CMDP like in HasanzadeZonuzy et al. [16]. Instead, we solve the empirical CMDP in Eq. (4) using specific primal-dual updates Eqs. (5) and (6). Note that if the rewards and constraint rewards (corresponding to $K$ constraints) are unknown and need to be estimated, a union bound guarantees that the sample complexity will only increase by a multiplicative $\log(K+1)$ factor [16].

In this setting, when using the *normalized value functions*, Bai et al. [6, Corollary 1] prove an $O\left(\frac{SA}{(1-\gamma)^2\varepsilon^2\zeta^2}\right)$ bound on the sample complexity. When translated to the standard $[0, 1/1-\gamma]$ range, this implies an $O\left(\frac{SA}{(1-\gamma)^4\varepsilon^2\zeta^2}\right)$ bound [6, Footnote 6]. In comparison, our result in Theorem 2 has a better dependence on $1/1-\gamma$ and does not depend on $\zeta$. Importantly, unlike [6], our result implies that in the relaxed feasibility setting, solving CMDPs is as hard as solving MDPs. In the next section, we instantiate Algorithm 1 in the strict feasibility setting.

## 5 Upper-bound under Strict Feasibility

Unlike Section 4, since the strict feasibility setting does not allow any constraint violations, it necessitates using a stricter constraint in the empirical CMDP to account for the estimation error in the transition probabilities. Algorithmically, we require setting $b' > b$. Specifically, in order to achieve the objective in Eq. (3) for a target error $\varepsilon > 0$, we require setting $N = \tilde{O}\left(\frac{\log(1/\delta)}{(1-\gamma)^5\zeta^2\varepsilon^2}\right)$,[6] $b' = b + \frac{\varepsilon(1-\gamma)\zeta}{20}$ and $\omega = \frac{\varepsilon(1-\gamma)}{10}$. This completely specifies the empirical CMDP $\hat{M}$ and the problem in Eq. (4). To specify the primal-dual algorithm, we set $U = \frac{4(1+\omega)}{\zeta(1-\gamma)}$, $\varepsilon_1 = O\left(\varepsilon^2(1-\gamma)^4\zeta^2\right)$ and $T = O\left(1/(1-\gamma)^6\zeta^4\varepsilon^2\right)$. With these choices, we prove the following theorem in Appendix C, and provide a proof sketch below.

**Theorem 3.** For a fixed $\varepsilon \in (0, 1/1-\gamma]$ and $\delta \in (0, 1)$, Algorithm 1, with $N = \tilde{O}\left(\frac{\log(1/\delta)}{(1-\gamma)^5\varepsilon^2\zeta^2}\right)$ samples, $b' = b + \frac{\varepsilon(1-\gamma)\zeta}{20}$, $\omega = \frac{\varepsilon(1-\gamma)}{10}$, $U = \frac{4(1+\omega)}{\zeta(1-\gamma)}$, $\varepsilon_1 = O\left(\varepsilon^2(1-\gamma)^4\zeta^2\right)$ and $T = O\left(1/(1-\gamma)^6\zeta^4\varepsilon^2\right)$ returns policy $\bar{\pi}_T$ that satisfies the objective in Eq. (3), with probability at least $1 - 4\delta$.

*Proof Sketch:* We prove the result for a general $b' = b + \Delta$ for $\Delta > 0$ and primal-dual error $\varepsilon_{\text{opt}} < \Delta$, and subsequently specify $\Delta$ (and hence $b'$) and $\varepsilon_{\text{opt}}$. In Lemma 12 (proved in Appendix C), we prove that if the constraint value functions are sufficiently concentrated (the empirical value function is close to the ground truth value function) for both the optimal policy $\pi^*$ in $M$ and the mixture policy $\bar{\pi}_T$ returned by Algorithm 1 i.e. if

$$\left|V_c^{\bar{\pi}_T}(\rho) - \hat{V}_c^{\bar{\pi}_T}(\rho)\right| \leq \Delta - \varepsilon_{\text{opt}} \quad ; \quad \left|V_c^{\pi^*}(\rho) - \hat{V}_c^{\pi^*}(\rho)\right| \leq \Delta \tag{10}$$

then (i) policy $\bar{\pi}_T$ satisfies the constraint in $M$ i.e. $V_c^{\bar{\pi}_T}(\rho) \geq b$, and (ii) its suboptimality in $M$ (compared to $\pi^*$) can be decomposed as:

$$V_r^{\pi^*}(\rho) - V_r^{\bar{\pi}_T}(\rho) \leq \frac{2\omega}{1-\gamma} + \varepsilon_{\text{opt}} + 2\Delta|\lambda^*| + \left|V_{r_p}^{\pi^*}(\rho) - \hat{V}_{r_p}^{\pi^*}(\rho)\right| + \left|\hat{V}_{r_p}^{\bar{\pi}_T}(\rho) - V_{r_p}^{\bar{\pi}_T}(\rho)\right| \tag{11}$$

In order to upper-bound $|\lambda^*|$, we require a concentration result for policy $\pi_c^* := \arg\max V_c^\pi(\rho)$ that maximizes the the constraint value function. In particular, we require $\Delta \in \left(0, \frac{\zeta}{2}\right)$ and $\left|V_c^{\pi_c^*}(\rho) - \hat{V}_c^{\pi_c^*}(\rho)\right| \leq \frac{\zeta}{2} - \Delta$. In Case 2 of Lemma 9 (proved in Appendix A), we show that if this concentration result holds, then we can upper-bound the optimal dual variable $|\lambda^*|$ by $\frac{2(1+\omega)}{\zeta(1-\gamma)}$. Using the above bounds to simplify Eq. (11),

$$V_r^{\pi^*}(\rho) - V_r^{\bar{\pi}_T}(\rho) \leq \frac{2\omega}{1-\gamma} + \varepsilon_{\text{opt}} + \frac{4\Delta(1+\omega)}{\zeta(1-\gamma)} + \left|V_{r_p}^{\pi^*}(\rho) - \hat{V}_{r_p}^{\pi^*}(\rho)\right| + \left|\hat{V}_{r_p}^{\bar{\pi}_T}(\rho) - V_{r_p}^{\bar{\pi}_T}(\rho)\right|.$$

With these results in hand, we can instantiate all the algorithm parameters except $N$ (the number of samples required for each state-action pair). In particular, we set $\Delta = \frac{\varepsilon(1-\gamma)\zeta}{40} < \frac{\zeta}{2}$, $\varepsilon_{\text{opt}} = \frac{\Delta}{5} = \frac{\varepsilon(1-\gamma)\zeta}{200} < \frac{\varepsilon}{5}$, and $\omega = \frac{\varepsilon(1-\gamma)}{10} < 1$. We set $U = \frac{8}{\zeta(1-\gamma)}$ for the primal-dual algorithm, ensuring that the $U > |\lambda^*|$ condition required by Theorem 1 holds. In order to guarantee that the primal-dual algorithm outputs an $\frac{\varepsilon(1-\gamma)\zeta}{200}$-approximate policy, we use Theorem 1 to set $T = O\left(\frac{1}{(1-\gamma)^6\zeta^4\varepsilon^2}\right)$

---

[6]Again, we do not need to know $\zeta$ and it can be replaced by the estimator constructed in Section F.

iterations and $\varepsilon_1 = O\left(\varepsilon^2(1-\gamma)^4\zeta^2\right)$. With these values, we can further simplify Eq. (11),

$$V_r^{\pi^*}(\rho) - V_r^{\bar{\pi}_T}(\rho) \leq \frac{3\varepsilon}{5} + \left|V_{r_p}^{\pi^*}(\rho) - \hat{V}_{r_p}^{\pi^*}(\rho)\right| + \left|\hat{V}_{r_p}^{\bar{\pi}_T}(\rho) - V_{r_p}^{\bar{\pi}_T}(\rho)\right|.$$

Putting everything together, in order to guarantee an $\varepsilon$-reward suboptimality for $\bar{\pi}_T$, we require the following concentration results to hold for $\Delta = \frac{\varepsilon(1-\gamma)\zeta}{40}$,

$$\left|V_c^{\bar{\pi}_T}(\rho) - \hat{V}_c^{\bar{\pi}_T}(\rho)\right| \leq \frac{4\Delta}{5} \; ; \; \left|V_c^{\pi^*}(\rho) - \hat{V}_c^{\pi^*}(\rho)\right| \leq \Delta \; ; \; \left|V_c^{\pi_c^*}(\rho) - \hat{V}_c^{\pi_c^*}(\rho)\right| \leq \frac{19\Delta}{5}$$

$$\left|V_{r_p}^{\pi^*}(\rho) - \hat{V}_{r_p}^{\pi^*}(\rho)\right| \leq \frac{\varepsilon}{5} \; ; \; \left|\hat{V}_{r_p}^{\bar{\pi}_T}(\rho) - V_{r_p}^{\bar{\pi}_T}(\rho)\right| \leq \frac{\varepsilon}{5}. \tag{12}$$

We control such concentration terms for both the constraint and reward value functions in Section 6, and bound the terms in Eq. (12). In particular, we prove that for a fixed $\varepsilon \in (0, 1/1-\gamma]$, using $N \geq \tilde{O}\left(\frac{\log(1/\delta)}{(1-\gamma)^5\zeta^2\varepsilon^2}\right)$ ensures that the statements in Eq. (12) hold with probability $1 - 4\delta$. This guarantees that $V_r^{\pi^*}(\rho) - V_r^{\bar{\pi}_T}(\rho) \leq \varepsilon$ and $V_c^{\bar{\pi}_T}(\rho) \geq b$. $\qquad\square$

Hence, the total sample-complexity of achieving the objective in Eq. (3) is $\tilde{O}\left(\frac{SA\log(1/\delta)}{(1-\gamma)^5\zeta^2\varepsilon^2}\right)$. Similar to Section 4, in the strict feasibility setting, with the *normalized value functions*, Bai et al. [6] prove an $O\left(\frac{SA}{(1-\gamma)^2\varepsilon^2\zeta^2}\right)$ bound on the sample complexity. When translated to the standard $[0, 1/1-\gamma]$ range, this implies an $\Omega\left(\frac{SA}{(1-\gamma)^6\varepsilon^2\zeta^2}\right)$ bound (see Appendix G for a detailed explanation). In comparison, our result in Theorem 3 has a better dependence on $(1/1-\gamma)$.

In Section 7, we prove a matching lower bound showing that Algorithm 1 is minimax optimal in the strict feasibility setting. In the next section, we give more details for the bounding the concentration terms in Theorem 2 and Theorem 3.

## 6    Bounding the concentration terms

We have seen that proving Theorem 2 and Theorem 3 require bounding the concentration terms in Eq. (9) and Eq. (12) respectively. In this section, we detail the techniques to achieve these bounds.

Our approach requires reasoning about a general unconstrained MDP $M_\alpha = (\mathcal{S}, \mathcal{A}, \mathcal{P}, \gamma, \alpha)$ with the same state-action space, transition probabilities and discount factor as the CMDP in Eq. (1) but with rewards equal to $\alpha$, coming from $[0, \alpha_{\max}]$. Analogously, we define the empirical MDP $\hat{M}_\alpha = (\mathcal{S}, \mathcal{A}, \hat{\mathcal{P}}, \gamma, \alpha)$ where the empirical transition matrix $\hat{\mathcal{P}}$ is the same as that of the empirical CMDP in Eq. (4). Similarly, we define MDP (and its empirical counterpart) $M_\beta = (\mathcal{S}, \mathcal{A}, \mathcal{P}, \gamma, \beta)$ (and $\hat{M}_\beta$) where the rewards $\beta$ are from $[0, \beta_{\max}]$. Note that the rewards $\alpha$ and $\beta$ are independent of the sampling of the transition matrix. The corresponding value functions for policy $\pi$ in $M_\alpha$ and $\hat{M}_\alpha$ (and $M_\beta$ and $\hat{M}_\beta$) are denoted as $V_\alpha^\pi$ and $\hat{V}_\alpha^\pi$ (and $V_\beta^\pi$ and $\hat{V}_\beta^\pi$) respectively, with the optimal value functions denoted as $V_\alpha^*$ and $\hat{V}_\alpha^*$ (and $V_\beta^*$ and $\hat{V}_\beta^*$) respectively. The action-value function in $M_\alpha$ for policy $\pi$ and state-action pair $(s, a)$ is denoted as $Q_\alpha^\pi(s, a)$ and analogously for $\hat{M}_\alpha$. For the subsequent technical results, we require that $\hat{M}_\alpha$ satisfy the following gap condition [21]:

**Definition 4** ($\iota$-Gap Condition). *MDP $\hat{M}_\alpha$ satisfies the $\iota$-gap condition if $\forall s$, $\hat{V}_\alpha^*(s) - \max_{a': a \neq \hat{\pi}_\alpha^*(s)} \hat{Q}_\alpha^*(s, a') \geq \iota$, where $\hat{\pi}_\alpha^* := \arg\max \hat{V}_\alpha^\pi$ and $\hat{\pi}_\alpha^*(s) = \arg\max_a \hat{Q}_\alpha^*(s, a)$ is the optimal action in state $s$.*

Intuitively, the gap condition states that there is a unique optimal action at each state and there is a gap between the performance of best action and the second best action. With this gap condition, we use techniques in Li et al. [21] to prove the following lemma in Appendix D.

**Lemma 5.** *Define $\hat{\pi}_\alpha^* := \arg\max_\pi \hat{V}_\alpha^\pi$. If (i) $\mathcal{E}$ is the event that the $\iota$-gap condition in Definition 4 holds for $\hat{M}_\alpha$ and (ii) for $\delta \in (0, 1)$ and $C(\delta) = 72 \log\left(\frac{16\alpha_{\max}SA\log(e/1-\gamma)}{(1-\gamma)^2\iota\delta}\right)$, the number of samples per state-action pair is $N \geq \frac{4C(\delta)}{1-\gamma}$, then with probability at least $\Pr[\mathcal{E}] - \delta/10$,*

$$\left\|\hat{V}_\beta^{\hat{\pi}_\alpha^*} - V_\beta^{\hat{\pi}_\alpha^*}\right\|_\infty \leq \sqrt{\frac{C(\delta)}{N \cdot (1-\gamma)^3}} \, \|\beta\|_\infty \, .$$

Hence, for policy $\hat{\pi}_\alpha^*$, we can obtain a concentration result in another MDP $\hat{M}_\beta$ with an independent reward function $\beta$ and the same empirical transition matrix $\hat{\mathcal{P}}$.

We wish to use the above lemma for the unconstrained MDP formed at every iteration of the primal update in Eq. (5). In particular, for a given $\lambda_t$, we will use Lemma 5 with $\alpha = r_p + \lambda_t c$ and $\beta = r_p$. Doing so will immediately give us a bound on $\left\| V_{r_p}^{\hat{\pi}_t} - \hat{V}_{r_p}^{\hat{\pi}_t} \right\|_\infty$ and hence $\left| V_{r_p}^{\hat{\pi}_t}(\rho) - \hat{V}_{r_p}^{\hat{\pi}_t}(\rho) \right|$. In order to use Lemma 5, we require the unconstrained MDP $\hat{M}_{r_p+\lambda c}$ to satisfy the gap condition in Definition 4 for any $\lambda \in \Lambda$. This is achieved by the perturbation of the rewards in Line 3 of Algorithm 1. Specifically, using Li et al. [21, Lemma 6] with a union-bound over $\Lambda$, we prove (in Lemma 16 in Appendix D) that with probability $1 - \delta/10$, $\hat{M}_{r_p+\lambda c}$ satisfies the gap condition in Definition 4 with $\iota = \frac{\omega\,\delta\,(1-\gamma)}{30\,|\Lambda||S||A|^2}$ for every $\lambda \in \Lambda$. This allows us to use Lemma 5 with $\alpha = r_p + \lambda_t c$ for all $t \in [T]$, and $\beta = r_p$ and $\beta = c$. In the following theorem, we obtain a concentration result for each $\hat{\pi}_t$ and hence for the mixture policy $\bar{\pi}_T$.

**Theorem 6.** For $\delta \in (0,1)$, $\omega \le 1$ and $C(\delta) = 72 \log \left( \frac{16(1+U+\omega)\,SA\log(e/1-\gamma)}{(1-\gamma)^2\,\iota\,\delta} \right)$ where $\iota = \frac{\omega\,\delta\,(1-\gamma)\,\varepsilon_1}{30\,U|S||A|^2}$, if $N \ge \frac{4\,C(\delta)}{1-\gamma}$, then for $\bar{\pi}_T$ output by Algorithm 1, with probability at least $1 - \delta/5$,

$$\left| V_{r_p}^{\bar{\pi}_T}(\rho) - \hat{V}_{r_p}^{\bar{\pi}_T}(\rho) \right| \le 2\sqrt{\frac{C(\delta)}{N \cdot (1-\gamma)^3}} \quad ; \quad \left| V_c^{\bar{\pi}_T}(\rho) - V_c^{\bar{\pi}_T}(\rho) \right| \le \sqrt{\frac{C(\delta)}{N \cdot (1-\gamma)^3}}.$$

Eqs. (9) and (12) also require proving concentration bounds for fixed (that do not depend on the data) policies $\pi^*$ and $\pi_c^*$. This can be done by directly using Li et al. [21, Lemma 1]. Specifically, we prove the following the lemma in Appendix D.

**Lemma 7.** For $\delta \in (0,1)$, $\omega \le 1$ and $C'(\delta) = 72 \log \left( \frac{4|S| \log(e/1-\gamma)}{\delta} \right)$, if $N \ge \frac{4\,C'(\delta)}{1-\gamma}$ and $B(\delta, N) := \sqrt{\frac{C'(\delta)}{(1-\gamma)^3 N}}$, then with probability at least $1 - 3\delta$,

$$\left| V_{r_p}^{\pi^*}(\rho) - \hat{V}_{r_p}^{\pi^*}(\rho) \right| \le 2B(\delta, N)\,; \quad \left| V_c^{\pi^*}(\rho) - \hat{V}_c^{\pi^*}(\rho) \right| \le B(\delta, N)\,; \quad \left| V_c^{\pi_c^*}(\rho) - \hat{V}_c^{\pi_c^*}(\rho) \right| \le B(\delta, N).$$

Using Theorem 6 and Lemma 7, we can bound each term in Eq. (9) and Eq. (12), completing the proof of Theorem 2 and Theorem 3 respectively. In the next section, we prove a lower-bound on the sample-complexity in the strict feasibility setting.

## 7 Lower-bound under strict feasibility

For a target error of $\varepsilon$, our lower bound construction demonstrates that it is important to estimate the constraint value function to a smaller error equal to $\varepsilon' := \varepsilon(1-\gamma)\zeta$. Intuitively, this is because a small ($\varepsilon'$) estimation error in the constraint value can incorrectly render the optimal policy infeasible and result in a large $\varepsilon'/(1-\gamma)\zeta$ suboptimality in the reward value. In Appendix E.1, we detail this intuition in a simplified bandit setting and present the formal CMDP lower-bound below.

We define an algorithm to be $(\varepsilon, \delta)$-sound if it outputs a policy $\hat{\pi}$ such that with probability $1 - \delta$, $V_r^*(\rho) - V_r^{\hat{\pi}}(\rho) \le \varepsilon$ and $V_c^{\hat{\pi}}(\rho) \ge b$ i.e. the algorithm achieves the strict feasibility objective in Eq. (3). We prove a lower bound on the number of samples required by any $(\varepsilon, \delta)$-sound algorithm on the CMDP instance in Fig. 1. For this instance, with a specific setting of the rewards and probabilities $p_0 < \bar{p} < p_1$, we prove that any $(\varepsilon, \delta)$-sound algorithm requires at least $\Omega \left( \frac{\ln(|S||A|/4\delta)}{\varepsilon^2 \zeta^2 (1-\gamma)^5} \right)$ samples to distinguish between $M_0$ and $M_{i,a}$. In particular, we prove the following theorem in Appendix E.2.

**Theorem 8.** There exists constants $\gamma_0 \in (1 - 1/\log(|S|), 1)$, $0 \le \varepsilon_0 \le \frac{1}{(1-\gamma)} \min \left\{ 1, \frac{\gamma}{(1-\gamma)\zeta} \right\}$, $\delta_0 \in (0,1)$, such that, for any $\gamma \in (\gamma_0, 1)$, $\varepsilon \in (0, \varepsilon_0), \delta \in (0, \delta_0)$, any $(\varepsilon, \delta)$-sound algorithm requires $\Omega \left( \frac{SA \ln(1/4\delta)}{\varepsilon^2 \zeta^2 (1-\gamma)^5} \right)$ samples from the generative model in the worst case.

The above lower bound matches the upper bound in Theorem 3 and proves that Algorithm 1 is near minimax optimal in the strict feasibility setting. It also demonstrates that solving CMDPs under strict

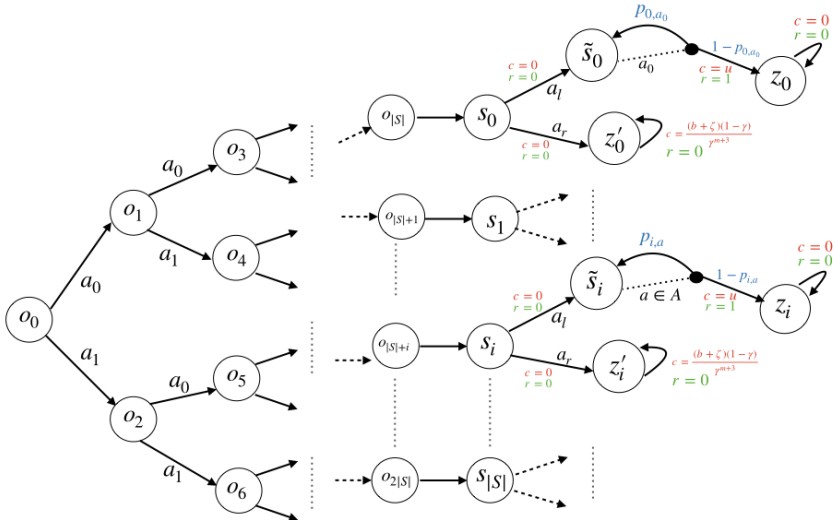

Figure 1: The instance consists of CMDPs with $S = 2^m - 1$ (for some integer $m > 0$) states and $A$ actions. We consider $SA + 1$ CMDPs – $M_0$ and $M_{i,a}$ ($i \in \{1, \ldots S\}$, $a \in \{1, \ldots, A\}$) that share the same structure shown in the figure. For each CMDP, $o_0$ is the fixed starting state and there is a deterministic path of length $m + 1$ from $o_0$ to each of the $S + 1$ states – $s_i$ (for $i \in \{0, 1, \ldots, S\}$). Except for states $\tilde{s}_i$, the transitions in all other states are deterministic. For $i \neq 0$, for action $a \in \mathcal{A}$ in state $\tilde{s}_i$, the probability of staying in $\tilde{s}_i$ is $p_{i,a}$, while that of transitioning to state $z_i$ is $1 - p_{i,a}$. There is only one action $a_0$ in $\tilde{s}_0$ and the probability of staying in $\tilde{s}_0$ is $p_{0,a_0}$, while that of transitioning to state $z_0$ is $1 - p_{0,a_0}$. The CMDPs $M_0$ and $M_{i,a}$ only differ in the values of $p_{i,a}$. The rewards $r$ and constraint rewards $c$ are the same in all CMDPs and are denoted in green and red respectively.

feasibility is inherently more difficult than solving unconstrained MDPs or CMDPs in the relaxed feasibility setting. Finally, we can conclude that the problem becomes more difficult (requires more samples) as the Slater constant $\zeta$ decreases and the feasible region shrinks.

## 8 Discussion

We proposed a model-based primal-dual algorithm for planning in CMDPs. Via upper and lower bounds, we proved that our algorithm is near minimax optimal for both the relaxed and strict feasibility settings. Our results demonstrate that solving CMDPs is as easy as MDPs when small constraint violations are allowed, but inherently more difficult when we demand zero constraint violation. Algorithmically, we required a specific primal-dual approach that involved solving a sequence of MDPs. In contrast, model-based approaches for MDPs [2, 21] allow the use of any black-box planner. It is possible to obtain an $O\left(\frac{S^2 A}{(1-\gamma)^3 \varepsilon^2}\right)$ sample complexity for a black-box CMDP planner in the relaxed feasibility setting. However, the $O(S^2 A)$ dependence in the bound implies that we need to accurately estimate all entries in the transition probability matrix, and is therefore loose in the special case of unconstrained MDPs [2, 21]. In the future, we aim to extend our near-optimal sample complexity results to black-box CMDP solvers.

## Acknowledgements

We would like to thank Reza Babanezhad and Arushi Jain for helpful feedback on the paper. Csaba Szepesvári gratefully acknowledges the funding from Natural Sciences and Engineering Research Council (NSERC) of Canada, "Design.R AI-assisted CPS Design" (DARPA) project and the Canada CIFAR AI Chairs Program for Amii. Lin Yang is supported in part by DARPA grant HR00112190130 and NSF Award 2221871. This work was partly done while Lin Yang was visiting Deepmind.

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
