*Proof.* We will define the dual regret w.r.t $\lambda$ as the following quantity, $R^d(\lambda, T)$.

$$R^d(\lambda, T) := \sum_{t=0}^{T-1} (\lambda_t - \lambda) (\hat{V}_c^{\hat{\pi}_t}(\rho) - b') \tag{13}$$

Using the primal update in Eq. (5), for any $\pi$,

$$\hat{V}_{r_p}^{\hat{\pi}_t}(\rho) + \lambda_t \hat{V}_c^{\hat{\pi}_t}(\rho) \geq \hat{V}_{r_p}^{\pi}(\rho) + \lambda_t \hat{V}_c^{\pi}(\rho) \tag{14}$$

Substituting $\pi = \hat{\pi}^*$, we have,

$$\hat{V}_{r_p}^{\hat{\pi}^*}(\rho) - \hat{V}_{r_p}^{\hat{\pi}_t}(\rho) \leq \lambda_t [\hat{V}_c^{\hat{\pi}_t}(\rho) - \hat{V}_c^{\hat{\pi}^*}(\rho)] \tag{15}$$

Since $\hat{\pi}^*$ is a solution to the empirical CMDP, $\hat{V}_c^{\hat{\pi}^*}(\rho) \geq b'$, we have

$$\hat{V}_{r_p}^{\hat{\pi}^*}(\rho) - \hat{V}_{r_p}^{\hat{\pi}_t}(\rho) \leq \lambda_t [\hat{V}_c^{\hat{\pi}_t}(\rho) - b'] \tag{16}$$

Averaging from $t = 0$ to $T - 1$, and using the definition of the dual regret in Eq. (13),

$$\frac{1}{T} \left[ \sum_{t=0}^{T-1} [\hat{V}_{r_p}^{\hat{\pi}^*}(\rho) - \hat{V}_{r_p}^{\hat{\pi}_t}(\rho)] \right] \leq \frac{R^d(0, T)}{T} \tag{17}$$

This allows us to get a handle on the average optimality gap in terms of the average dual regret. For the second part, starting from the definition of the dual regret,

$$\sum_{t=0}^{T-1} (\lambda_t - \lambda) (\hat{V}_c^{\hat{\pi}_t}(\rho) - b') = R^d(\lambda, T)$$

Dividing by $T$, and using Eq. (16),

$$\frac{1}{T} \sum_{t=0}^{T-1} \left[ \hat{V}_{r_p}^{\hat{\pi}^*}(\rho) - \hat{V}_{r_p}^{\hat{\pi}_t}(\rho) \right] + \frac{\lambda}{T} \underbrace{\sum_{t=0}^{T-1} (b' - \hat{V}_c^{\hat{\pi}_t}(\rho))}_{:=A} \leq \frac{R^d(\lambda, T)}{T}$$

The above statement holds for all $\lambda$. If $A \leq 0$, it implies that $\frac{1}{T}\left[\sum_{t=0}^{T-1}(b' - \hat{V}_c^{\hat{\pi}_t}(\rho))\right] \leq 0$, which is the desired bound on the constraint violation. In this case, we can set $\lambda = 0$ and this recovers Eq. (17).

If $A > 0$, we will set $\lambda = U$. We define $\bar{\pi}_T = \frac{1}{T}\sum_{t=0}^{T-1}\hat{\pi}_t$ as a mixture policy. Using the linearity of $\hat{V}_{r_p}^\pi(\rho)$ and $\hat{V}_c^\pi(\rho)$, i.e. $\frac{1}{T}\sum_{t=0}^{T-1}\hat{V}_{r_p}^{\hat{\pi}_t}(\rho) = \hat{V}_{r_p}^{\bar{\pi}_T}(\rho)$, and $\frac{1}{T}\sum_{t=0}^{T-1}\hat{V}_c^{\hat{\pi}_t}(\rho) = \hat{V}_c^{\bar{\pi}_T}(\rho)$, we obtain the following equation,

$$\left[\hat{V}_{r_p}^{\hat{\pi}^*}(\rho) - \hat{V}_{r_p}^{\bar{\pi}_T}(\rho)\right] + U\left(b' - \hat{V}_c^{\bar{\pi}_T}(\rho)\right) \leq R^d(U, T)$$

Since $(b' - \hat{V}_c^{\bar{\pi}_T}(\rho)) > 0$,

$$\left[\hat{V}_{r_p}^{\hat{\pi}^*}(\rho) - \hat{V}_{r_p}^{\bar{\pi}_T}(\rho)\right] + U\left[b' - \hat{V}_c^{\bar{\pi}_T}(\rho)\right]_+ \leq \frac{R^d(U, T)}{T}$$

where, $[x]_+ = \max\{x, 0\}$

Using Lemma 10 for $U > \lambda^*$,

$$\left[b' - \hat{V}_c^{\bar{\pi}_T}(\rho)\right]_+ \leq \frac{R^d(U, T)}{T(U - \lambda^*)} \tag{18}$$

It remains to bound the dual regret for the gradient descent procedure for updating $\lambda_t$. For this, we bound the distance $|\lambda_{t+1} - \lambda|$ for a general $\lambda \in [0, U]$. Defining $\lambda'_{t+1} := \mathbb{P}_{[0,U]}[\lambda_t - \eta\left(\hat{V}_c^{\hat{\pi}_t}(\rho) - b'\right)]$,

$$|\lambda_{t+1} - \lambda| = |\mathcal{R}_\Lambda[\lambda'_{t+1}] - \lambda| = |\mathcal{R}_\Lambda[\lambda'_{t+1}] - \lambda'_{t+1} + \lambda'_{t+1} - \lambda| \leq |\mathcal{R}_\Lambda[\lambda'_{t+1}] - \lambda'_{t+1}| + |\lambda'_{t+1} - \lambda|$$
$$\leq \varepsilon_l + |\lambda'_{t+1} - \lambda|$$
(Since $|\lambda - \mathcal{R}_\Lambda[\lambda]| \leq \varepsilon_l$ for all $\lambda \in [0, U]$ because of the epsilon-net.)

Squaring both sides,

$$|\lambda_{t+1} - \lambda|^2 = \varepsilon_l^2 + |\lambda'_{t+1} - \lambda|^2 + 2\varepsilon_l |\lambda'_{t+1} - \lambda| \leq \varepsilon_l^2 + 2\varepsilon_l U + |\lambda'_{t+1} - \lambda|^2$$
(Since $\lambda, \lambda'_{t+1} \in [0, U]$,)
$$\leq \varepsilon_l^2 + 2\varepsilon_l U + |\lambda_t - \eta\left(\hat{V}_c^{\hat{\pi}_t}(\rho) - b'\right) - \lambda|^2$$
(Since projections are non-expansive)
$$= \varepsilon_l^2 + 2\varepsilon_l U + |\lambda_t - \lambda|^2 - 2\eta\left(\lambda_t - \lambda\right)\left(\hat{V}_c^{\hat{\pi}_t}(\rho) - b'\right) + \eta^2(\hat{V}_c^{\hat{\pi}_t}(\rho) - b')^2$$
$$\leq \varepsilon_l^2 + 2\varepsilon_l U + |\lambda_t - \lambda|^2 - 2\eta\left(\lambda_t - \lambda\right)\left(\hat{V}_c^{\hat{\pi}_t}(\rho) - b'\right) + \frac{\eta^2}{(1-\gamma)^2}$$

Rearranging and dividing by $2\eta$,

$$\left(\lambda_t - \lambda\right)\left(\hat{V}_c^{\hat{\pi}_t}(\rho) - b'\right) \leq \frac{\varepsilon_l^2 + 2\varepsilon_l U}{2\eta} + \frac{|\lambda_t - \lambda|^2 - |\lambda_{t+1} - \lambda|^2}{2\eta} + \frac{\eta}{2(1-\gamma)^2}$$

Summing from $t = 0$ to $T - 1$ and using the definition of the dual regret,

$$R^d(\lambda, T) \leq T\frac{\varepsilon_l^2 + 2\varepsilon_l U}{2\eta} + \frac{1}{2\eta}\sum_{t=0}^{T-1}[|\lambda_t - \lambda|^2 - |\lambda_{t+1} - \lambda|^2] + \frac{\eta T}{2(1-\gamma)^2}$$

Telescoping and bounding $|\lambda_0 - \lambda|$ by $U$,

$$\leq T\frac{\varepsilon_l^2 + 2\varepsilon_l U}{2\eta} + \frac{U^2}{2\eta} + \frac{\eta T}{2(1-\gamma)^2}$$

Setting $\eta = \frac{U(1-\gamma)}{\sqrt{T}}$,

$$R^d(\lambda, T) \leq T^{3/2}\frac{\varepsilon_l^2 + 2\varepsilon_l U}{2U(1-\gamma)} + \frac{U\sqrt{T}}{1-\gamma} \tag{19}$$

Using Eq. (19) with the expressions for the optimality gap (Eq. (17)) and constraint violation (Eq. (18)), we obtain the following bounds.

For the reward optimality gap, since $\lambda = 0 \in [0, U]$,

$$\hat{V}_{r_p}^{\hat{\pi}^*}(\rho) - \hat{V}_{r_p}^{\bar{\pi}_T}(\rho) \leq \sqrt{T}\frac{\varepsilon_1^2 + 2\varepsilon_1 U}{2U(1-\gamma)} + \frac{U}{(1-\gamma)\sqrt{T}} < \sqrt{T}\frac{3\varepsilon_1}{2(1-\gamma)} + \frac{U}{(1-\gamma)\sqrt{T}}$$
$$\text{(Since } \varepsilon_1 < U\text{)}$$

For the constraint violation, since $U \in [0, U]$,

$$\left[b' - \hat{V}_c^{\bar{\pi}_T}(\rho)\right] \leq \left[b' - \hat{V}_c^{\bar{\pi}_T}(\rho)\right]_+ \leq \sqrt{T}\frac{\varepsilon_1^2 + 2\varepsilon_1 U}{2U(1-\gamma)(U-\lambda^*)} + \frac{U}{(U-\lambda^*)(1-\gamma)\sqrt{T}}$$
$$< \sqrt{T}\frac{3\varepsilon_1}{2(1-\gamma)(U-\lambda^*)} + \frac{U}{(U-\lambda^*)(1-\gamma)\sqrt{T}}$$
$$\text{(Since } \varepsilon_1 < U\text{)}$$

Let us set $T$ s.t. that the second term in both quantities is bound by $\frac{\varepsilon_{\text{opt}}}{2}$,

$$T = T_0 := \frac{4U^2}{\varepsilon_{\text{opt}}^2(1-\gamma)^2}\left[1 + \frac{1}{(U-\lambda^*)^2}\right]$$

With $T = T_0$, the above expressions can be simplified as:

$$\hat{V}_{r_p}^{\hat{\pi}^*}(\rho) - \hat{V}_{r_p}^{\bar{\pi}_T}(\rho) \leq \frac{2U}{(1-\gamma)\varepsilon_{\text{opt}}}\left(1 + \frac{1}{U-\lambda^*}\right)\frac{3\varepsilon_1}{2(1-\gamma)} + \frac{\varepsilon_{\text{opt}}}{2}$$

$$\left[b' - \hat{V}_c^{\bar{\pi}_T}(\rho)\right] \leq \frac{2U}{(1-\gamma)\varepsilon_{\text{opt}}}\left(1 + \frac{1}{U-\lambda^*}\right)\frac{3\varepsilon_1}{2(1-\gamma)(U-\lambda^*)} + \frac{\varepsilon_{\text{opt}}}{2}$$

We want to set $\varepsilon_1$ s.t. the first term in both quantities is also bounded by $\frac{\varepsilon_{\text{opt}}}{2}$,

$$\varepsilon_1 = \frac{\varepsilon_{\text{opt}}^2(1-\gamma)^2(U-\lambda^*)}{6U}$$

With these values, the algorithm ensures that,

$$\hat{V}_{r_p}^{\hat{\pi}^*}(\rho) - \hat{V}_{r_p}^{\bar{\pi}_T}(\rho) \leq \varepsilon_{\text{opt}} \quad ; \quad \left[b' - \hat{V}_c^{\bar{\pi}_T}(\rho)\right] \leq \varepsilon_{\text{opt}}.$$

$\square$

---

**Lemma 9** (Bounding the dual variable). *The objective Eq. (4) satisfies strong duality. Defining $\pi_c^* := \arg\max V_c^\pi(\rho)$. We consider two cases: (1) If $b' = b - \varepsilon'$ for $\varepsilon' > 0$ and event $\mathcal{E}_1 = \left\{\left|\hat{V}_c^{\pi_c^*}(\rho) - V_c^{\pi_c^*}(\rho)\right| \leq \frac{\varepsilon'}{2}\right\}$ holds, then $\lambda^* \leq \frac{2(1+\omega)}{\varepsilon'(1-\gamma)}$ and (2) If $b' = b + \Delta$ for $\Delta \in \left(0, \frac{\varsigma}{2}\right)$ and event $\mathcal{E}_2 = \left\{\left|\hat{V}_c^{\pi_c^*}(\rho) - V_c^{\pi_c^*}(\rho)\right| \leq \frac{\varsigma}{2} - \Delta\right\}$ holds, then $\lambda^* \leq \frac{2(1+\omega)}{\varsigma(1-\gamma)}$.*

---

*Proof.* Writing the empirical CMDP in Eq. (4) in its Lagrangian form,

$$\hat{V}_{r_p}^{\hat{\pi}^*}(\rho) = \max_\pi \min_{\lambda \geq 0} \hat{V}_{r_p}^\pi(\rho) + \lambda[\hat{V}_c^\pi(\rho) - b']$$

Using the linear programming formulation of CMDPs in terms of the state-occupancy measures $\mu$, we know that both the objective and the constraint are linear functions of $\mu$, and strong duality holds w.r.t $\mu$. Since $\mu$ and $\pi$ have a one-one mapping, we can switch the min and the max [24], implying,

$$= \min_{\lambda \geq 0} \max_\pi \hat{V}_{r_p}^\pi(\rho) + \lambda[\hat{V}_c^\pi(\rho) - b']$$

Since $\lambda^*$ is the optimal dual variable for the empirical CMDP in Eq. (4),

$$= \max_\pi \hat{V}_{r_p}^\pi(\rho) + \lambda^*[\hat{V}_c^\pi(\rho) - b']$$

Define $\pi_c^* := \arg\max V_c^\pi(\rho)$ and $\hat{\pi}_c^* := \arg\max \hat{V}_c^\pi(\rho)$

$$\geq \hat{V}_{r_p}^{\hat{\pi}_c^*}(\rho) + \lambda^* [\hat{V}_c^{\hat{\pi}_c^*}(\rho) - b']$$
$$= \hat{V}_{r_p}^{\hat{\pi}_c^*}(\rho) + \lambda^* \left[\left(\hat{V}_c^{\hat{\pi}_c^*}(\rho) - V_c^{\pi_c^*}(\rho)\right) + (V_c^{\pi_c^*}(\rho) - b) + (b - b')\right]$$

By definition, $\zeta = V_c^{\pi_c^*}(\rho) - b$

$$= \hat{V}_{r_p}^{\hat{\pi}_c^*}(\rho) + \lambda^* \left[\left(\hat{V}_c^{\hat{\pi}_c^*}(\rho) - \hat{V}_c^{\pi_c^*}(\rho)\right) + \left(\hat{V}_c^{\pi_c^*}(\rho) - V_c^{\pi_c^*}(\rho)\right) + \zeta + (b - b')\right]$$

By definition of $\hat{\pi}_c^*$, $\left(\hat{V}_c^{\hat{\pi}_c^*}(\rho) - \hat{V}_c^{\pi_c^*}(\rho)\right) \geq 0$

$$\hat{V}_{r_p}^{\hat{\pi}^*}(\rho) \geq \hat{V}_{r_p}^{\hat{\pi}_c^*}(\rho) + \lambda^* \left[\zeta + (b - b') - \left|\hat{V}_c^{\pi_c^*}(\rho) - V_c^{\pi_c^*}(\rho)\right|\right]$$

1) If $b' = b - \varepsilon'$ for $\varepsilon' > 0$. Hence,

$$\hat{V}_{r_p}^{\hat{\pi}^*}(\rho) \geq \hat{V}_{r_p}^{\hat{\pi}_c^*}(\rho) + \lambda^* \left[\zeta + \varepsilon' - \left|\hat{V}_c^{\pi_c^*}(\rho) - V_c^{\pi_c^*}(\rho)\right|\right]$$

If the event $\mathcal{E}_1$ holds, $\left|\hat{V}_c^{\pi_c^*}(\rho) - V_c^{\pi_c^*}(\rho)\right| \leq \frac{\varepsilon'}{2}$, implying, $\left|\hat{V}_c^{\pi_c^*}(\rho) - V_c^{\pi_c^*}(\rho)\right| < \zeta + \frac{\varepsilon'}{2}$, then,

$$\geq \hat{V}_{r_p}^{\hat{\pi}_c^*}(\rho) + \lambda^* \frac{\varepsilon'}{2}$$
$$\implies \lambda^* \leq \frac{2}{\varepsilon'}[\hat{V}_{r_p}^{\hat{\pi}^*}(\rho) - \hat{V}_{r_p}^{\hat{\pi}_c^*}(\rho)] \leq \frac{2(1+\omega)}{\varepsilon'(1-\gamma)}$$

2) If $b' = b + \Delta$ for $\Delta \in \left(0, \frac{\zeta}{2}\right)$. Hence,

$$\hat{V}_{r_p}^{\hat{\pi}^*}(\rho) \geq \hat{V}_{r_p}^{\hat{\pi}_c^*}(\rho) + \lambda^* \left[\zeta - \Delta - \left|\hat{V}_c^{\pi_c^*}(\rho) - V_c^{\pi_c^*}(\rho)\right|\right]$$

If the event $\mathcal{E}_2$ holds, $\left|\hat{V}_c^{\pi_c^*}(\rho) - V_c^{\pi_c^*}(\rho)\right| \leq \frac{\zeta}{2} - \Delta$ for $\Delta < \frac{\zeta}{2}$, then,

$$\geq \hat{V}_{r_p}^{\hat{\pi}_c^*}(\rho) + \lambda^* \frac{\zeta}{2}$$
$$\implies \lambda^* \leq \frac{2}{\zeta}[\hat{V}_{r_p}^{\hat{\pi}^*}(\rho) - \hat{V}_{r_p}^{\hat{\pi}_c^*}(\rho)] \leq \frac{2(1+\omega)}{\zeta(1-\gamma)}$$

$\square$

---

**Lemma 10** (Lemma B.2 of Jain et al. [17]). *For any $C > \lambda^*$ and any $\tilde{\pi}$ s.t. $\hat{V}_{r_p}^{\hat{\pi}^*}(\rho) - \hat{V}_{r_p}^{\tilde{\pi}}(\rho) + C[b - \hat{V}_c^{\tilde{\pi}}(\rho)]_+ \leq \beta$, we have $[b - \hat{V}_c^{\tilde{\pi}}(\rho)]_+ \leq \frac{\beta}{C - \lambda^*}$.*

---

*Proof.* Define $\nu(\tau) = \max_\pi\{V_r^\pi(\rho) \mid V_c^\pi(\rho) \geq b + \tau\}$ and note that by definition, $\nu(0) = V_r^{\hat{\pi}^*}(\rho)$ and that $\nu$ is a decreasing function for its argument.

Let $V_l^{\pi,\lambda}(\rho) = V_r^\pi(\rho) + \lambda(V_c^\pi(\rho) - b)$. Then, for any policy $\pi$ s.t. $V_c^\pi(\rho) \geq b + \tau$, we have

$$V_l^{\pi,\lambda^*}(\rho) \leq \max_{\pi'} V_l^{\pi',\lambda^*}(\rho)$$
$$= V_r^{\hat{\pi}^*}(\rho) \qquad \text{(by strong duality)}$$
$$= \nu(0) \qquad \text{(from above relation)}$$
$$\implies \nu(0) - \tau\lambda^* \geq V_l^{\pi,\lambda^*}(\rho) - \tau\lambda^* = V_r^\pi(\rho) + \lambda^* \underbrace{(V_c^\pi(\rho) - b - \tau)}_{\text{Non-negative}}$$
$$\implies \nu(0) - \tau\lambda^* \geq \max_\pi\{V_r^\pi(\rho) \mid V_c^\pi(\rho) \geq b + \tau\} = \nu(\tau).$$
$$\implies \tau\lambda^* \leq \nu(0) - \nu(\tau). \tag{20}$$

Now we choose $\tilde{\tau} = -(b - V_c^{\tilde{\pi}}(\rho))_+$.

$$
\begin{aligned}
(C - \lambda^*)|\tilde{\tau}| &= \lambda^* \tilde{\tau} + C|\tilde{\tau}| && \text{(since } \tilde{\tau} \leq 0) \\
&\leq \nu(0) - \nu(\tilde{\tau}) + C|\tilde{\tau}| && \text{(Eq. (20))} \\
&= V_r^{\hat{\pi}^*}(\rho) - V_r^{\tilde{\pi}}(\rho) + C|\tilde{\tau}| + V_r^{\tilde{\pi}}(\rho) - \nu(\tilde{\tau}) && \text{(definition of } \nu(0)) \\
&= V_r^{\hat{\pi}^*}(\rho) - V_r^{\tilde{\pi}}(\rho) + C(b - V_c^{\tilde{\pi}}(\rho))_+ + V_r^{\tilde{\pi}}(\rho) - \nu(\tilde{\tau}) \\
&\leq \beta + V_r^{\tilde{\pi}}(\rho) - \nu(\tilde{\tau}) .
\end{aligned}
$$

Now let us bound $\nu(\tilde{\tau})$:

$$
\begin{aligned}
\nu(\tilde{\tau}) &= \max_\pi \{V_r^\pi(\rho) \mid V_c^\pi(\rho) \geq b - (b - V_c^{\tilde{\pi}}(\rho))_+\} \\
&\geq \max_\pi \{V_r^\pi(\rho) \mid V_c^\pi(\rho) \geq V_c^{\tilde{\pi}}(\rho)\} && \text{(tightening the constraint)}
\end{aligned}
$$

$$
\nu(\tilde{\tau}) \geq V_r^{\tilde{\pi}}(\rho) \implies (C - \lambda^*)|\tilde{\tau}| \leq \beta \implies (b - V_c^{\tilde{\pi}}(\rho))_+ \leq \frac{\beta}{C - \lambda^*}
$$

$\square$

## B   Proof of Theorem 2

**Theorem 2.** For a fixed $\varepsilon \in (0, 1/1-\gamma]$ and $\delta \in (0, 1)$, Algorithm 1 with $N = \tilde{O}\left(\frac{\log(1/\delta)}{(1-\gamma)^3 \varepsilon^2}\right)$ samples, $b' = b - \frac{3\varepsilon}{8}, \omega = \frac{\varepsilon(1-\gamma)}{8}, U = O\left(1/\varepsilon(1-\gamma)\right), \varepsilon_1 = O\left(\varepsilon^2(1-\gamma)^2\right)$ and $T = O\left(1/(1-\gamma)^4 \varepsilon^4\right)$, returns policy $\bar{\pi}_T$ that satisfies the objective in Eq. (2) with probability at least $1 - 4\delta$.

*Proof.* We fill in the details required for the proof sketch in the main paper. Proceeding according to the proof sketch, we first detail the computation of $T$ and $\varepsilon_1$ for the primal-dual algorithm. Recall that $U = \frac{32}{5\varepsilon(1-\gamma)}$ and $\varepsilon_{\text{opt}} = \frac{\varepsilon}{4}$. Using Theorem 1, we need to set

$$
T = \frac{4U^2}{\varepsilon_{\text{opt}}^2 (1-\gamma)^2}\left[1 + \frac{1}{(U - \lambda^*)^2}\right] = \frac{64}{\varepsilon^2(1-\gamma)^2}\left[1 + \frac{1}{(U - \lambda^*)^2}\right]
$$

Recall that $|\lambda^*| \leq C := \frac{16}{5\varepsilon(1-\gamma)}$ and $U = 2C$. Simplifying,

$$
\leq \frac{256}{\varepsilon^2(1-\gamma)^2}\left[C^2 + 1\right] < \frac{512}{\varepsilon^2(1-\gamma)^2}C^2 = \frac{512}{\varepsilon^2(1-\gamma)^2}\frac{256}{25\varepsilon^2(1-\gamma)^2}
$$

$$
\implies T = O\left(1/\varepsilon^4(1-\gamma)^4\right).
$$

Using Theorem 1, we need to set $\varepsilon_1$,

$$
\varepsilon_1 = \frac{\varepsilon_{\text{opt}}^2(1-\gamma)^2 (U - \lambda^*)}{6U} = \frac{\varepsilon^2(1-\gamma)^2 (U - \lambda^*)}{96U} \leq \frac{\varepsilon^2(1-\gamma)^2}{96}
$$

$$
\implies \varepsilon_1 = O\left(\varepsilon^2(1-\gamma)^2\right).
$$

For bounding the concentration terms for $\bar{\pi}_T$ in Eq. (9), we use Theorem 6 with $U = \frac{32}{5\varepsilon(1-\gamma)}$, $\omega = \frac{\varepsilon(1-\gamma)}{8}$ and $\varepsilon_1 = \frac{\varepsilon^2(1-\gamma)^2}{96}$. In this case, $\iota = \frac{\omega \delta (1-\gamma) \varepsilon_1}{30 U |S||A|^2} = O\left(\frac{\delta \varepsilon^4 (1-\gamma)^4}{SA^2}\right)$ and

$$
C(\delta) = 72 \log\left(\frac{16(1 + U + \omega) SA \log(e/1-\gamma)}{(1-\gamma)^2 \iota \delta}\right) = O\left(\log\left(\frac{S^2 A^3}{\delta^2 \varepsilon^5 (1-\gamma)^7}\right)\right).
$$

With this value of $C(\delta)$, in order to satisfy the concentration bounds for $\bar{\pi}_T$, we require that

$$
2\sqrt{\frac{C(\delta)}{N \cdot (1-\gamma)^3}} \leq \frac{\varepsilon}{4} \implies N \geq O\left(\frac{C(\delta)}{(1-\gamma)^3 \varepsilon^2}\right)
$$

We use the Lemma 7 to bound the remaining concentration terms for $\pi^*$ and $\pi_c^*$ in Eq. (9). In this case, for $C'(\delta) = 72 \log\left(\frac{4S \log(e/1-\gamma)}{\delta}\right)$, we require that,

$$
2\sqrt{\frac{C'(\delta)}{N \cdot (1-\gamma)^3}} \leq \frac{\varepsilon}{4} \implies N \geq O\left(\frac{C'(\delta)}{(1-\gamma)^3 \varepsilon^2}\right)
$$

Hence, if $N \geq \tilde{O}\left(\frac{\log(1/\delta)}{(1-\gamma)^3 \varepsilon^2}\right)$, the bounds in Eq. (9) are satisfied, completing the proof. $\square$

**Lemma 11** (Decomposing the suboptimality). *For $b' = b - \frac{\varepsilon - \varepsilon_{opt}}{2}$, if (i) $\varepsilon_{opt} < \varepsilon$, and (ii) the following conditions are satisfied,*

$$\left| V_c^{\bar{\pi}_T}(\rho) - \hat{V}_c^{\bar{\pi}_T}(\rho) \right| \leq \frac{\varepsilon - \varepsilon_{opt}}{2} \; ; \; \left| V_c^{\pi^*}(\rho) - \hat{V}_c^{\pi^*}(\rho) \right| \leq \frac{\varepsilon - \varepsilon_{opt}}{2}$$

*where $\pi_c^* := \arg \max V_c^{\pi}(\rho)$, then (a) policy $\bar{\pi}_T$ violates the constraint by at most $\varepsilon$ i.e. $V_c^{\bar{\pi}_T}(\rho) \geq b - \varepsilon$ and (b) its optimality gap can be bounded as:*

$$V_r^{\pi^*}(\rho) - V_r^{\bar{\pi}_T}(\rho) \leq \frac{2\omega}{1 - \gamma} + \varepsilon_{opt} + \left| V_{r_p}^{\pi^*}(\rho) - \hat{V}_{r_p}^{\pi^*}(\rho) \right| + \left| \hat{V}_{r_p}^{\bar{\pi}_T}(\rho) - V_{r_p}^{\bar{\pi}_T}(\rho) \right|$$

*Proof.* From Theorem 1, we know that,

$$\hat{V}_c^{\bar{\pi}_T}(\rho) \geq b' - \varepsilon_{\text{opt}} \implies V_c^{\bar{\pi}_T}(\rho) \geq V_c^{\bar{\pi}_T}(\rho) - \hat{V}_c^{\bar{\pi}_T}(\rho) + b' - \varepsilon_{\text{opt}} \geq - \left| V_c^{\bar{\pi}_T}(\rho) - \hat{V}_c^{\bar{\pi}_T}(\rho) \right| + b' - \varepsilon_{\text{opt}}$$

Since we require $\bar{\pi}_T$ to violate the constraint in the true CMDP by at most $\varepsilon$, we require $V_c^{\bar{\pi}_T}(\rho) \geq b - \varepsilon$. From the above equation, a sufficient condition for ensuring this is,

$$- \left| V_c^{\bar{\pi}_T}(\rho) - \hat{V}_c^{\bar{\pi}_T}(\rho) \right| + b' - \varepsilon_{\text{opt}} \geq b - \varepsilon \,,$$

meaning that we require

$$\left| V_c^{\bar{\pi}_T}(\rho) - \hat{V}_c^{\bar{\pi}_T}(\rho) \right| \leq (b' - b) - \varepsilon_{\text{opt}} + \varepsilon .$$

Plugging in the value of $b'$, we see that this sufficient condition indeed holds, by our assumption that $\left| V_c^{\bar{\pi}_T}(\rho) - \hat{V}_c^{\bar{\pi}_T}(\rho) \right| \leq \frac{\varepsilon - \varepsilon_{\text{opt}}}{2}$.

Let $\pi^*$ be the solution to Eq. (1). Our next goal is to show that $\pi^*$ is feasible for the constrained problem in Eq. (4), i.e., $\hat{V}_c^{\pi^*}(\rho) \geq b'$. We have

$$V_c^{\pi^*}(\rho) \geq b \implies \hat{V}_c^{\pi^*}(\rho) \geq b - \left| V_c^{\pi^*}(\rho) - \hat{V}_c^{\pi^*}(\rho) \right|$$

Since we require $\hat{V}_c^{\pi^*}(\rho) \geq b'$, using the above equation, a sufficient condition to ensure this is

$$b - \left| V_c^{\pi^*}(\rho) - \hat{V}_c^{\pi^*}(\rho) \right| \geq b' \text{ meaning that we require } \left| V_c^{\pi^*}(\rho) - \hat{V}_c^{\pi^*}(\rho) \right| \leq b - b'.$$

Since $b' = b - \frac{\varepsilon - \varepsilon_{\text{opt}}}{2}$, we require that

$$\left| V_c^{\pi^*}(\rho) - \hat{V}_c^{\pi^*}(\rho) \right| \leq \frac{\varepsilon - \varepsilon_{\text{opt}}}{2}.$$

Given that the above statements hold, we can decompose the suboptimality in the reward value function as follows:

$$V_r^{\pi^*}(\rho) - V_r^{\bar{\pi}_T}(\rho)$$

$$= V_r^{\pi^*}(\rho) - V_{r_p}^{\pi^*}(\rho) + V_{r_p}^{\pi^*}(\rho) - V_r^{\bar{\pi}_T}(\rho)$$

$$= [V_r^{\pi^*}(\rho) - V_{r_p}^{\pi^*}(\rho)] + V_{r_p}^{\pi^*}(\rho) - \hat{V}_{r_p}^{\pi^*}(\rho) + \hat{V}_{r_p}^{\pi^*}(\rho) - V_r^{\bar{\pi}_T}(\rho)$$

$$\leq [V_r^{\pi^*}(\rho) - V_{r_p}^{\pi^*}(\rho)] + [V_{r_p}^{\pi^*}(\rho) - \hat{V}_{r_p}^{\pi^*}(\rho)] + \hat{V}_{r_p}^{\hat{\pi}^*}(\rho) - V_r^{\bar{\pi}_T}(\rho)$$

$$\quad \text{(By optimality of } \hat{\pi}^* \text{ and since we have ensured that } \pi^* \text{ is feasible for Eq. (4))}$$

$$= [V_r^{\pi^*}(\rho) - V_{r_p}^{\pi^*}(\rho)] + [V_{r_p}^{\pi^*}(\rho) - \hat{V}_{r_p}^{\pi^*}(\rho)] + [\hat{V}_{r_p}^{\hat{\pi}^*}(\rho) - \hat{V}_{r_p}^{\bar{\pi}_T}(\rho)] + \hat{V}_{r_p}^{\bar{\pi}_T}(\rho) - V_r^{\bar{\pi}_T}(\rho)$$

$$= \underbrace{[V_r^{\pi^*}(\rho) - V_{r_p}^{\pi^*}(\rho)]}_{\text{Perturbation Error}} + \underbrace{[V_{r_p}^{\pi^*}(\rho) - \hat{V}_{r_p}^{\pi^*}(\rho)]}_{\text{Concentration Error}} + \underbrace{[\hat{V}_{r_p}^{\hat{\pi}^*}(\rho) - \hat{V}_{r_p}^{\bar{\pi}_T}(\rho)]}_{\text{Primal-Dual Error}} + \underbrace{[\hat{V}_{r_p}^{\bar{\pi}_T}(\rho) - V_{r_p}^{\bar{\pi}_T}(\rho)]}_{\text{Concentration Error}} + \underbrace{[V_{r_p}^{\bar{\pi}_T}(\rho) - V_r^{\bar{\pi}_T}(\rho)]}_{\text{Perturbation Error}}$$

For a perturbation magnitude equal to $\omega$, we use Lemma 15 to bound both perturbation errors by $\frac{\omega}{1-\gamma}$. Using Theorem 1 to bound the primal-dual error by $\varepsilon_{\text{opt}}$,

$$V_r^{\pi^*}(\rho) - V_r^{\bar{\pi}_T}(\rho) \leq \frac{2\omega}{1 - \gamma} + \varepsilon_{\text{opt}} + \underbrace{[V_{r_p}^{\pi^*}(\rho) - \hat{V}_{r_p}^{\pi^*}(\rho)]}_{\text{Concentration Error}} + \underbrace{[\hat{V}_{r_p}^{\bar{\pi}_T}(\rho) - V_{r_p}^{\bar{\pi}_T}(\rho)]}_{\text{Concentration Error}} .$$

$\square$

# C  Proof of Theorem 3

**Theorem 3.** *For a fixed* $\varepsilon \in (0, 1/1-\gamma]$ *and* $\delta \in (0, 1)$*, Algorithm 1, with* $N = \tilde{O}\left(\frac{\log(1/\delta)}{(1-\gamma)^5\varepsilon^2\zeta^2}\right)$ *samples,* $b' = b + \frac{\varepsilon(1-\gamma)\zeta}{20}$*,* $\omega = \frac{\varepsilon(1-\gamma)}{10}$*,* $U = \frac{4(1+\omega)}{\zeta(1-\gamma)}$*,* $\varepsilon_l = O\left(\varepsilon^2(1-\gamma)^4\zeta^2\right)$ *and* $T = O\left(1/(1-\gamma)^6\zeta^4\varepsilon^2\right)$ *returns policy* $\bar{\pi}_T$ *that satisfies the objective in Eq. (3), with probability at least* $1 - 4\delta$*.*

*Proof.* We fill in the details required for the proof sketch in the main paper. Proceeding according to the proof sketch, we first detail the computation of $T$ and $\varepsilon_l$ for the primal-dual algorithm. Recall that $U = \frac{8}{\zeta(1-\gamma)}$, $\Delta = \frac{\varepsilon(1-\gamma)\zeta}{40}$ and $\varepsilon_{\text{opt}} = \frac{\Delta}{5}$. Using Theorem 1, we need to set

$$T = \frac{4U^2}{\varepsilon_{\text{opt}}^2(1-\gamma)^2}\left[1 + \frac{1}{(U-\lambda^*)^2}\right] = \frac{100}{\Delta^2(1-\gamma)^2}\left[1 + \frac{1}{(U-\lambda^*)^2}\right]$$

Recall that $|\lambda^*| \leq C := \frac{4}{\zeta(1-\gamma)}$ and $U = 2C$. Simplifying,

$$\leq \frac{400}{\Delta^2(1-\gamma)^2}\left[C^2 + 1\right] < \frac{800}{\Delta^2(1-\gamma)^2}C^2 = \frac{800}{\Delta^2(1-\gamma)^2}\frac{16}{\zeta^2(1-\gamma)^2}$$

$$\implies T \leq \frac{800 \cdot 1600}{\varepsilon^2\zeta^2(1-\gamma)^4}\frac{16}{\zeta^2(1-\gamma)^2} = O\left(1/\varepsilon^2\zeta^4(1-\gamma)^6\right).$$

Using Theorem 1, we need to set $\varepsilon_l$,

$$\varepsilon_l = \frac{\varepsilon_{\text{opt}}^2(1-\gamma)^2(U-\lambda^*)}{6U} = \frac{\Delta^2(1-\gamma)^2(U-\lambda^*)}{150U} \leq \frac{\Delta^2(1-\gamma)^2}{150}$$

$$\implies \varepsilon_l \leq \frac{\varepsilon^2\zeta^2(1-\gamma)^4}{150 \cdot 1600} = O\left(\varepsilon^2\zeta^2(1-\gamma)^4\right).$$

For bounding the concentration terms for $\bar{\pi}_T$ in Eq. (12), we use Theorem 6 with $U = \frac{8}{\zeta(1-\gamma)}$, $\omega = \frac{\varepsilon(1-\gamma)}{10}$ and $\varepsilon_l = \frac{\varepsilon^2\zeta^2(1-\gamma)^4}{150\cdot1600}$. In this case, $\iota = \frac{\omega\,\delta\,(1-\gamma)\,\varepsilon_l}{30\,U|S||A|^2} = O\left(\frac{\delta\varepsilon^3\zeta^3(1-\gamma)^7}{SA^2}\right)$ and

$$C(\delta) = 72\log\left(\frac{16(1+U+\omega)\,SA\log\left(e/1-\gamma\right)}{(1-\gamma)^2\,\iota\,\delta}\right) = O\left(\log\left(\frac{S^2A^3}{(1-\gamma)^{10}\delta^2\varepsilon^3\zeta^4}\right)\right).$$

With this value of $C(\delta)$, in order to satisfy the concentration bounds for $\bar{\pi}_T$, we require that

$$2\sqrt{\frac{C(\delta)}{N\cdot(1-\gamma)^3}} \leq \frac{\Delta}{5} \implies N \geq O\left(\frac{C(\delta)}{(1-\gamma)^3\,\Delta^2}\right) \geq O\left(\frac{C(\delta)}{(1-\gamma)^5\,\zeta^2\,\varepsilon^2}\right)$$

We use the Lemma 7 to bound the remaining concentration terms for $\pi^*$ and $\pi_c^*$ in Eq. (12). In this case, for $C'(\delta) = 72\log\left(\frac{4S\log(e/1-\gamma)}{\delta}\right)$, we require that,

$$2\sqrt{\frac{C'(\delta)}{N\cdot(1-\gamma)^3}} \leq \frac{\Delta}{5} \implies N \geq O\left(\frac{C'(\delta)}{(1-\gamma)^3\,\Delta^2}\right) \geq O\left(\frac{C'(\delta)}{(1-\gamma)^5\,\zeta^2\,\varepsilon^2}\right)$$

Hence, if $N \geq \tilde{O}\left(\frac{\log(1/\delta)}{(1-\gamma)^5\,\zeta^2\,\varepsilon^2}\right)$, the bounds in Eq. (12) are satisfied, completing the proof. $\qquad\square$

**Lemma 12** (Decomposing the suboptimality). *For a fixed $\Delta > 0$ and $\varepsilon_{opt} < \Delta$, if $b' = b + \Delta$, then the following conditions are satisfied,*

$$\left| V_c^{\bar{\pi}_T}(\rho) - \hat{V}_c^{\bar{\pi}_T}(\rho) \right| \leq \Delta - \varepsilon_{opt} \, ; \, \left| V_c^{\pi^*}(\rho) - \hat{V}_c^{\pi^*}(\rho) \right| \leq \Delta$$

*then (a) policy $\bar{\pi}_T$ satisfies the constraint i.e. $V_c^{\bar{\pi}_T}(\rho) \geq b$ and (b) its optimality gap can be bounded as:*

$$V_r^{\pi^*}(\rho) - V_r^{\bar{\pi}_T}(\rho) \leq \frac{2\omega}{1-\gamma} + \varepsilon_{opt} + 2\Delta\lambda^* + \left| V_{r_p}^{\pi^*}(\rho) - \hat{V}_{r_p}^{\pi^*}(\rho) \right| + \left| \hat{V}_{r_p}^{\bar{\pi}_T}(\rho) - V_{r_p}^{\bar{\pi}_T}(\rho) \right|.$$

*Proof.* Compared to Eq. (4), we define a slightly modified CMDP problem by changing the constraint RHS to $b''$ for some $b''$ to be specified later. We denote its corresponding optimal policy as $\tilde{\pi}^*$. In particular,

$$\tilde{\pi}^* \in \arg\max_{\pi} \hat{V}_{r_p}^{\pi}(\rho) \text{ s.t. } \hat{V}_c^{\pi}(\rho) \geq b'' \tag{21}$$

From Theorem 1, we know that,

$$\hat{V}_c^{\bar{\pi}_T}(\rho) \geq b' - \varepsilon_{\text{opt}} \implies V_c^{\bar{\pi}_T}(\rho) \geq V_c^{\bar{\pi}_T}(\rho) - \hat{V}_c^{\bar{\pi}_T}(\rho) + b' - \varepsilon_{\text{opt}} \geq - \left| V_c^{\bar{\pi}_T}(\rho) - \hat{V}_c^{\bar{\pi}_T}(\rho) \right| + b' - \varepsilon_{\text{opt}}$$

Since we require $\bar{\pi}_T$ to satisfy the constraint in the true CMDP, we require $V_c^{\bar{\pi}_T}(\rho) \geq b$. From the above equation, a sufficient condition for ensuring this is,

$$- \left| V_c^{\bar{\pi}_T}(\rho) - \hat{V}_c^{\bar{\pi}_T}(\rho) \right| + b' - \varepsilon_{\text{opt}} \geq b$$

meaning that we require $\left| V_c^{\bar{\pi}_T}(\rho) - \hat{V}_c^{\bar{\pi}_T}(\rho) \right| \leq (b' - b) - \varepsilon_{\text{opt}}$.

In the subsequent analysis, we will require $\pi^*$ to be feasible for the constrained problem in Eq. (21). This implies that we require $\hat{V}_c^{\pi^*}(\rho) \geq b''$. Since $\pi^*$ is the solution to Eq. (1), we know that,

$$V_c^{\pi^*}(\rho) \geq b \implies \hat{V}_c^{\pi^*}(\rho) \geq b - \left| V_c^{\pi^*}(\rho) - \hat{V}_c^{\pi^*}(\rho) \right|$$

Since we require $\hat{V}_c^{\pi^*}(\rho) \geq b''$, using the above equation, a sufficient condition to ensure this is

$$b - \left| V_c^{\pi^*}(\rho) - \hat{V}_c^{\pi^*}(\rho) \right| \geq b'' \text{ meaning that we require } \left| V_c^{\pi^*}(\rho) - \hat{V}_c^{\pi^*}(\rho) \right| \leq b - b''.$$

Hence we require the following statements to hold:

$$\left| V_c^{\bar{\pi}_T}(\rho) - \hat{V}_c^{\bar{\pi}_T}(\rho) \right| \leq (b' - b) - \varepsilon_{\text{opt}} \quad ; \quad \left| V_c^{\pi^*}(\rho) - \hat{V}_c^{\pi^*}(\rho) \right| \leq b - b''.$$

Given that the above statements hold, we can decompose the suboptimality in the reward value function as follows:

$$V_r^{\pi^*}(\rho) - V_r^{\bar{\pi}_T}(\rho) = V_r^{\pi^*}(\rho) - V_{r_p}^{\pi^*}(\rho) + V_{r_p}^{\pi^*}(\rho) - V_r^{\bar{\pi}_T}(\rho)$$

$$= [V_r^{\pi^*}(\rho) - V_{r_p}^{\pi^*}(\rho)] + [V_{r_p}^{\pi^*}(\rho) - \hat{V}_{r_p}^{\pi^*}(\rho)] + \hat{V}_{r_p}^{\pi^*}(\rho) - V_r^{\bar{\pi}_T}(\rho)$$

$$\leq [V_r^{\pi^*}(\rho) - V_{r_p}^{\pi^*}(\rho)] + [V_{r_p}^{\pi^*}(\rho) - \hat{V}_{r_p}^{\pi^*}(\rho)] + \hat{V}_{r_p}^{\tilde{\pi}^*}(\rho) - V_r^{\bar{\pi}_T}(\rho)$$
$$\text{(By optimality of } \hat{\pi}^* \text{ and since we have ensured that } \pi^* \text{ is feasible for Eq. (21))}$$

$$= [V_r^{\pi^*}(\rho) - V_{r_p}^{\pi^*}(\rho)] + [V_{r_p}^{\pi^*}(\rho) - \hat{V}_{r_p}^{\pi^*}(\rho)] + [\hat{V}_{r_p}^{\tilde{\pi}^*}(\rho) - \hat{V}_{r_p}^{\hat{\pi}^*}(\rho)] + \hat{V}_{r_p}^{\hat{\pi}^*}(\rho) - V_r^{\bar{\pi}_T}(\rho)$$

$$= [V_r^{\pi^*}(\rho) - V_{r_p}^{\pi^*}(\rho)] + [V_{r_p}^{\pi^*}(\rho) - \hat{V}_{r_p}^{\pi^*}(\rho)] + [\hat{V}_{r_p}^{\tilde{\pi}^*}(\rho) - \hat{V}_{r_p}^{\hat{\pi}^*}(\rho)] + \hat{V}_{r_p}^{\hat{\pi}^*}(\rho) - V_r^{\bar{\pi}_T}(\rho)$$

$$= [V_r^{\pi^*}(\rho) - V_{r_p}^{\pi^*}(\rho)] + [V_{r_p}^{\pi^*}(\rho) - \hat{V}_{r_p}^{\pi^*}(\rho)] + [\hat{V}_{r_p}^{\tilde{\pi}^*}(\rho) - \hat{V}_{r_p}^{\hat{\pi}^*}(\rho)] + [\hat{V}_{r_p}^{\hat{\pi}^*}(\rho) - \hat{V}_{r_p}^{\bar{\pi}_T}(\rho)]$$
$$+ \hat{V}_{r_p}^{\bar{\pi}_T}(\rho) - V_r^{\bar{\pi}_T}(\rho)$$

$$= \underbrace{[V_r^{\pi^*}(\rho) - V_{r_p}^{\pi^*}(\rho)]}_{\text{Perturbation Error}} + \underbrace{[V_{r_p}^{\pi^*}(\rho) - \hat{V}_{r_p}^{\pi^*}(\rho)]}_{\text{Concentration Error}} + \underbrace{[\hat{V}_{r_p}^{\tilde{\pi}^*}(\rho) - \hat{V}_{r_p}^{\hat{\pi}^*}(\rho)]}_{\text{Sensitivity Error}} + \underbrace{[\hat{V}_{r_p}^{\hat{\pi}^*}(\rho) - \hat{V}_{r_p}^{\bar{\pi}_T}(\rho)]}_{\text{Primal-Dual Error}}$$

$$+ \underbrace{[\hat{V}_{r_p}^{\bar{\pi}_T}(\rho) - V_{r_p}^{\bar{\pi}_T}(\rho)]}_{\text{Concentration Error}} + \underbrace{[V_{r_p}^{\bar{\pi}_T}(\rho) - V_r^{\bar{\pi}_T}(\rho)]}_{\text{Perturbation Error}}$$

For a perturbation magnitude equal to $\omega$, we use Lemma 15 to bound both perturbation errors by $\frac{\omega}{1-\gamma}$. Using Theorem 1 to bound the primal-dual error by $\varepsilon_{\text{opt}}$,

$$\leq \frac{2\omega}{1-\gamma} + \varepsilon_{\text{opt}} + \underbrace{[V_{r_p}^{\pi^*}(\rho) - \hat{V}_{r_p}^{\pi^*}(\rho)]}_{\text{Concentration Error}} + \underbrace{[\hat{V}_{r_p}^{\tilde{\pi}^*}(\rho) - \hat{V}_{r_p}^{\hat{\pi}^*}(\rho)]}_{\text{Sensitivity Error}} + \underbrace{[\hat{V}_{r_p}^{\bar{\pi}_T}(\rho) - V_{r_p}^{\bar{\pi}_T}(\rho)]}_{\text{Concentration Error}}$$

Since $b' = b + \Delta$ and setting $b'' = b - \Delta$, we use Lemma 13 to bound the sensitivity error term,

$$V_r^{\pi^*}(\rho) - V_r^{\bar{\pi}_T}(\rho) \leq \frac{2\omega}{1-\gamma} + \varepsilon_{\text{opt}} + 2\Delta\lambda^* + \underbrace{[V_{r_p}^{\pi^*}(\rho) - \hat{V}_{r_p}^{\pi^*}(\rho)]}_{\text{Concentration Error}} + \underbrace{[\hat{V}_{r_p}^{\bar{\pi}_T}(\rho) - V_{r_p}^{\bar{\pi}_T}(\rho)]}_{\text{Concentration Error}}$$

With these values of $b'$ and $b''$, we require the following statements to hold,

$$\left| V_c^{\bar{\pi}_T}(\rho) - \hat{V}_c^{\bar{\pi}_T}(\rho) \right| \leq \Delta - \varepsilon_{\text{opt}} \quad ; \quad \left| V_c^{\pi^*}(\rho) - \hat{V}_c^{\pi^*}(\rho) \right| \leq \Delta.$$

$\square$

---

**Lemma 13** (Bounding the sensitivity error). *If $b' = b + \Delta$ and $b'' = b - \Delta$ in Eq. (4) and Eq. (21) such that,*

$$\hat{\pi}^* \in \arg \max_{\pi} \hat{V}_{r_p}^{\pi}(\rho) \text{ s.t. } \hat{V}_c^{\pi}(\rho) \geq b + \Delta$$

$$\tilde{\pi}^* \in \arg \max_{\pi} \hat{V}_{r_p}^{\pi}(\rho) \text{ s.t. } \hat{V}_c^{\pi}(\rho) \geq b - \Delta,$$

*then the sensitivity error term can be bounded by:*

$$\left| \hat{V}_{r_p}^{\hat{\pi}^*}(\rho) - \hat{V}_{r_p}^{\tilde{\pi}^*}(\rho) \right| \leq 2\Delta\lambda^*.$$

---

*Proof.* Writing the empirical CMDP in Eq. (4) in its Lagrangian form,

$$\hat{V}_{r_p}^{\hat{\pi}^*}(\rho) = \max_{\pi} \min_{\lambda \geq 0} \hat{V}_{r_p}^{\pi}(\rho) + \lambda[\hat{V}_c^{\pi}(\rho) - (b + \Delta)]$$

$$= \min_{\lambda \geq 0} \max_{\pi} \hat{V}_{r_p}^{\pi}(\rho) + \lambda[\hat{V}_c^{\pi}(\rho) - (b + \Delta)]$$

(By strong duality Lemma 9)

Since $\lambda^*$ is the optimal dual variable for the empirical CMDP in Eq. (4),

$$= \max_{\pi} \hat{V}_{r_p}^{\pi}(\rho) + \lambda^* [\hat{V}_c^{\pi}(\rho) - (b + \Delta)]$$

$$\geq \hat{V}_{r_p}^{\tilde{\pi}^*}(\rho) + \lambda^* [\hat{V}_c^{\tilde{\pi}^*}(\rho) - (b + \Delta)] \quad \text{(The relation holds for } \pi = \tilde{\pi}^*.)$$

Since $\hat{V}_c^{\tilde{\pi}^*}(\rho) \geq b - \Delta$,

$$\hat{V}_{r_p}^{\hat{\pi}^*}(\rho) \geq \hat{V}_{r_p}^{\tilde{\pi}^*}(\rho) - 2\lambda^*\Delta$$

$$\implies \hat{V}_{r_p}^{\tilde{\pi}^*}(\rho) - \hat{V}_{r_p}^{\hat{\pi}^*}(\rho) \leq 2\Delta\lambda^*$$

Since the CMDP in Eq. (21) (with $b'' = b - \Delta$) is a less constrained problem than the one in Eq. (4) (with $b' = b + \Delta$), $\hat{V}_{r_p}^{\tilde{\pi}^*}(\rho) \geq \hat{V}_{r_p}^{\hat{\pi}^*}(\rho)$, and hence,

$$\left| \hat{V}_{r_p}^{\tilde{\pi}^*}(\rho) - \hat{V}_{r_p}^{\hat{\pi}^*}(\rho) \right| \leq 2\Delta\lambda^*.$$

$\square$

# D  Concentration proofs

> **Lemma 5.** *Define $\hat{\pi}_\alpha^* := \arg\max_\pi \hat{V}_\alpha^\pi$. If (i) $\mathcal{E}$ is the event that the $\iota$-gap condition in Definition 4 holds for $\hat{M}_\alpha$ and (ii) for $\delta \in (0,1)$ and $C(\delta) = 72\log\left(\frac{16\alpha_{\max}SA\log(e/1-\gamma)}{(1-\gamma)^2\,\iota\,\delta}\right)$, the number of samples per state-action pair is $N \geq \frac{4\,C(\delta)}{1-\gamma}$, then with probability at least $\Pr[\mathcal{E}] - \delta/10$,*
> $$\left\| \hat{V}_\beta^{\hat{\pi}_\alpha^*} - V_\beta^{\hat{\pi}_\alpha^*} \right\|_\infty \leq \sqrt{\frac{C(\delta)}{N\cdot(1-\gamma)^3}}\,\|\beta\|_\infty .$$

*Proof.* Since the policy $\hat{\pi}_\alpha^*$ depends on the sampling, we can not directly apply the standard concentration results to bound $\left\| \hat{V}_\beta^{\hat{\pi}_\alpha^*} - \hat{V}_\beta^{\hat{\pi}_\alpha^*} \right\|_\infty$. We thus seek to apply a critical lemma established in Li et al. [21]. It begins with introducing a sequence of vectors for a general data-dependent policy $\pi$ and reward $\beta$, defined recursively as

$$V_\beta^{\pi,(0)} = (I - \gamma P^\pi)^{-1}\beta^\pi \quad \text{and} \quad V_\beta^{\pi,(l)} = (I - \gamma P^\pi)^{-1}\sqrt{\mathrm{Var}_{P_\pi}(V_\beta^{\pi,(l-1)})},\ \forall l \geq 1.$$

In their Lemma 2 (restated below), they show that if certain concentration relation can be established between the empirical and ground truth MDP, then $\left\| \hat{V}_\beta^\pi - \hat{V}_\beta^\pi \right\|_\infty$ can be bounded.

**Lemma 14** (Lemma 2 of [21]). *For a data-dependent policy $\pi$, suppose there exists a $\nu_1 \geq 0$ such that $\{V_\beta^{\pi,(l)}\}$ obeys*

$$\left|(\hat{\mathcal{P}}_\pi - \mathcal{P}_\pi)V_\beta^{\pi,(l)}\right| \leq \sqrt{\frac{\nu_1}{N}}\sqrt{\mathrm{Var}_{\mathcal{P}_\pi}[V_\beta^{\pi,(l)}]} + \frac{\nu_1\left\|V_\beta^{\pi,(l)}\right\|_\infty}{N},\ \textit{for all } 0 \leq l \leq \log\left(\frac{e}{1-\gamma}\right).$$

*Suppose that $N \geq \frac{16e^2}{1-\gamma}\nu_1$. Then*

$$\left\| \hat{V}_\beta^\pi - V_\beta^\pi \right\|_\infty \leq \frac{6}{1-\gamma}\cdot\sqrt{\frac{\nu_1}{N(1-\gamma)}}\,\|\beta\|_\infty .$$

To use this lemma for $\pi = \hat{\pi}_\alpha^*$, we will need to establish Bernstein-type bounds on $(\mathcal{P}_{s,a} - \hat{\mathcal{P}}_{s,a})V_\beta^{\hat{\pi}_\alpha^*,(l)}$ for all $(s,a)$ and integer $0 \leq l \leq \log(e/(1-\gamma))$. Since $\hat{\pi}_\alpha^*$ depends on the sampling, a direct concentration bound is not possible. Instead, we will first bound $(\mathcal{P}_{s,a} - \hat{\mathcal{P}}_{s,a})V_\beta^{\pi,(l)}$ for all $\pi \in \Pi_{s,a}$, where $\Pi_{s,a}$ is a random set independent of $\hat{\mathcal{P}}$, and then show that $\hat{\pi}^* \in \Pi_{s,a}$ with good probability.

First, we describe the construction of $\Pi_{s,a}$. We will follow the ideas in Agarwal et al. [2] and Li et al. [21], and construct an absorbing empirical MDP $\hat{M}_{s,a}$, which is the same as the original empirical MDP, but state-action pair $(s,a)$ is absorbing, i.e., $\hat{\mathcal{P}}'(s'|s,a) = 1$ if and only if $s' = s$. The reward for $(s,a)$ is equal to $u$. We define $\hat{V}_{s,a,\alpha,u}^\pi$ and $\hat{Q}_{s,a,\alpha,u}^\pi$ to be the value function and $Q$-function of policy $\pi$ for $\hat{M}_{s,a}$ with reward function $\alpha$, and define $\hat{\pi}_{s,a,\alpha,u}^*$ to be the optimal policy i.e. $\hat{\pi}_{s,a,\alpha,u}^* = \arg\max \hat{V}_{s,a,\alpha,u}^\pi$. We will use the shorthand $-\hat{V}_{\alpha,u}^* := \max \hat{V}_{s,a,\alpha,u}^\pi$ and $\hat{Q}_{\alpha,u}^* := \max \hat{Q}_{s,a,\alpha,u}^\pi$.

We consider a grid,

$$U_{s,a} = \{0, \pm\iota(1-\gamma)/2, \pm2\iota(1-\gamma)/2, \pm3\iota(1-\gamma)/2\ldots, \pm\alpha_{\max}\},$$

and define $\Pi_{s,a} = \{\hat{\pi}_{s,a,\alpha,u}^* : u \in U_{s,a}\}$. Then $\Pi_{s,a}$ is a random set independent of $\hat{\mathcal{P}}_{s,a}$. Let $L = \{0, 1, \ldots, \lceil\log(e/(1-\gamma))\rceil\}$. Then, by Lemma 19, we have, with probability at least $1 - \delta/|S|/|A|$, for all $\pi \in \Pi_{s,a}$ and $l \in L$

$$\left|(\mathcal{P}_{s,a} - \hat{\mathcal{P}}_{s,a})\cdot V_\beta^{\pi,(l)}\right| \leq \sqrt{\frac{2\log(4|U_{s,a}||L||S||A|/\delta)}{N}}\sqrt{\mathrm{Var}_{P_{s,a}}(V_\beta^{\pi,(l)})} + \frac{2\log(4|U_{s,a}||S||A||L|/\delta)\left\|V_\beta^{\pi,(l)}\right\|_\infty}{3N},$$

which we denote as event $\mathcal{E}_{s,a}$.

Next, we show that if $u^* = \hat{Q}_\alpha^*(s,a) - \gamma\hat{V}_\alpha^*(s)$, then $\hat{\pi}_{s,a,\alpha,u^*}^* = \hat{\pi}_\alpha^*$:

(1) If $\pi^*(s) = a$, it straightforward to verify that

$$\hat{V}_\alpha^*(s) = \hat{Q}_\alpha^*(s,a) = u^* + \gamma\hat{V}_\alpha^*(s) \geq r(s,a') + \hat{\mathcal{P}}(\cdot|s',a')^\top\hat{V}_\alpha^*, \quad \forall a' \neq a.$$

(2) If $\pi^*(s) \neq a$, then

$$\hat{V}_\alpha^*(s) = \max_{a'}\hat{Q}_\alpha^*(s,a') = \max_{a'}(r(s,a') + \hat{\mathcal{P}}(\cdot|s,a')^\top\hat{V}_\alpha^*) = r(s,\hat{\pi}^*(s)) + \hat{\mathcal{P}}(\cdot|s,\hat{\pi}^*(s))^\top\hat{V}_\alpha^*.$$

(3) For $s' \neq s$, we have

$$\hat{V}_\alpha^*(s') = \hat{Q}_\alpha^*(s',\hat{\pi}^*(s')) = r(s',\hat{\pi}^*(s')) + \hat{\mathcal{P}}(\cdot|s',\hat{\pi}^*(s'))^\top\hat{V}_\alpha^* = \max_{a'}(r(s,a') + \hat{\mathcal{P}}(\cdot|s',a')^\top\hat{V}_\alpha^*).$$

Therefore, $\hat{Q}_\alpha^*(s',a')$ and $\hat{\pi}_\alpha^*$ satisfies the Bellman equations in the absorbing MDP; consequently, we have $\hat{Q}_{s,a,\alpha,u}^{\hat{\pi}_\alpha^*} = \hat{Q}_{\alpha,u}^* = \hat{Q}_\alpha^*$ and $\hat{V}_{s,a,\alpha,u}^{\hat{\pi}_\alpha^*} = \hat{V}_{\alpha,u}^* = \hat{V}_\alpha^{\hat{\pi}_\alpha^*}$ and $\hat{\pi}_\alpha^*$ is an optimal policy in the absorbing MDP.

Moreover, suppose event $\mathcal{E}$ happens, then $\hat{Q}_\alpha^*$ satisfies the $\iota$-gap condition. By Lemma 17, for all $|u - u^*| \leq \iota(1-\gamma)/2$, we have

$$\hat{\pi}_\alpha^* = \hat{\pi}_{s,a,\alpha,u^*}^* = \hat{\pi}_{s,a,\alpha,u}^*.$$

Thus, if $\mathcal{E}$ happens, then $\hat{\pi}_\alpha^* = \hat{\pi}_{s,a,\alpha,u}^*$ for some $u \in U_{s,a}$ and thus $\hat{\pi}^* \in \Pi_{s,a}$. On $\mathcal{E} \cap \mathcal{E}_{s,a}$, we have, for all $l \in L$,

$$\left|(\mathcal{P}_{s,a} - \hat{\mathcal{P}}_{s,a}) \cdot V_\beta^{\hat{\pi}_\alpha^*,(l)}\right| \leq \sqrt{\frac{2\log(4|U_{s,a}||L||S||A|/\delta)}{N}}\sqrt{\mathrm{Var}_{P_{s,a}}(V_\beta^{\hat{\pi}_\alpha^*,(l)})} + \frac{2\log(4|U_{s,a}||S||A||L|/\delta)\left\|V_\beta^{\hat{\pi}_\alpha^*,(l)}\right\|_\infty}{3N}.$$

By a union bound over all $(s,a)$, we have, with probability at least $\Pr[\mathcal{E} \cap (\cap_{s,a}\mathcal{E}_{s,a})] \geq \Pr[\mathcal{E}] - \delta$, for all $(s,a)$, and $l \in L$,

$$\left|(\mathcal{P}_{s,a} - \hat{\mathcal{P}}_{s,a}) \cdot V_\beta^{\hat{\pi}_\alpha^*,(l)}\right| \leq \sqrt{\frac{2\log(4|U_{s,a}||L||S||A|/\delta)}{N}}\sqrt{\mathrm{Var}_{P_{s,a}}(V_\beta^{\hat{\pi}_\alpha^*,(l)})} + \frac{2\log(4|U_{s,a}||S||A||L|/\delta)\left\|V_\beta^{\hat{\pi}_\alpha^*,(l)}\right\|_\infty}{3N}.$$

Let $\nu_1 = 2\log(4|U_{s,a}||S||A||L|/\delta)$ and apply Lemma 14, we arrive at,

$$\left\|\hat{V}_\beta^{\hat{\pi}^*} - V_\beta^{\hat{\pi}^*}\right\|_\infty \leq \frac{6}{1-\gamma} \cdot \sqrt{\frac{\nu_1}{N(1-\gamma)}}\|\beta\|_\infty$$

provided $N \geq \frac{16e^2}{1-\gamma}\nu_1$. For instantiating $\nu_1$, note that $|U_{s,a}| = \frac{4\alpha_{\max}}{(1-\gamma)^2\iota}$, $|L| = \log(e/1-\gamma)$. Hence, $\nu_1 = 2\log\left(\frac{16\alpha_{\max}SA\log(e/1-\gamma)}{(1-\gamma)^2\iota\delta}\right)$.

$\square$

---

**Lemma 15.** *For any policy $\pi$, we have*

$$\left\|V_r^\pi(\rho) - V_{r_p}^\pi(\rho)\right\|_\infty \leq \frac{\omega}{1-\gamma} \quad ; \quad \left\|\hat{V}_r^\pi(\rho) - \hat{V}_{r_p}^\pi(\rho)\right\|_\infty \leq \frac{\omega}{1-\gamma}$$

---

*Proof.* For policy $\pi$, $V_r^\pi(\rho) = (I - \gamma P_\pi)^{-1}r^\pi$ and $V_{r_p}^\pi(\rho) = (I - \gamma P_\pi)^{-1}r_p^\pi$.

$$V_r^\pi(\rho) - V_{r_p}^\pi(\rho) = (I - \gamma P_\pi)^{-1}[r^\pi - r_p^\pi]$$

$$\implies \left\|V_r^\pi(\rho) - V_{r_p}^\pi(\rho)\right\|_\infty \leq \|(I - \gamma P_\pi)^{-1}\|_1\left\|r^\pi - r_p^\pi\right\|_\infty$$

Since $\|(I - \gamma P_\pi)^{-1}\|_1 \leq \frac{1}{1-\gamma}$ and $\left\|r^\pi - r_p^\pi\right\|_\infty \leq \omega$

$$\left\|V_r^\pi(\rho) - V_{r_p}^\pi(\rho)\right\|_\infty \leq \frac{\omega}{1-\gamma}.$$

The same argument can be used to bound $\left\|\hat{V}_r^\pi(\rho) - \hat{V}_{r_p}^\pi(\rho)\right\|_\infty$ completing the proof. $\square$

**Theorem 6.** For $\delta \in (0,1)$, $\omega \leq 1$ and $C(\delta) = 72 \log \left( \frac{16(1+U+\omega)\,SA \log(e/1-\gamma)}{(1-\gamma)^2\,\iota\,\delta} \right)$ where $\iota = \frac{\omega\,\delta\,(1-\gamma)\,\varepsilon_l}{30\,U|S||A|^2}$, if $N \geq \frac{4\,C(\delta)}{1-\gamma}$, then for $\bar{\pi}_T$ output by Algorithm 1, with probability at least $1 - \delta/5$,

$$\left| V_{r_p}^{\bar{\pi}_T}(\rho) - \hat{V}_{r_p}^{\bar{\pi}_T}(\rho) \right| \leq 2\sqrt{\frac{C(\delta)}{N \cdot (1-\gamma)^3}} \quad ; \quad \left| V_c^{\bar{\pi}_T}(\rho) - V_c^{\bar{\pi}_T}(\rho) \right| \leq \sqrt{\frac{C(\delta)}{N \cdot (1-\gamma)^3}}.$$

*Proof.*

$$\left| V_{r_p}^{\bar{\pi}_T}(\rho) - \hat{V}_{r_p}^{\bar{\pi}_T}(\rho) \right| = \left| \frac{1}{T} \sum_{t=0}^{T-1} \left[ V_{r_p}^{\pi_t}(\rho) - \hat{V}_{r_p}^{\pi_t}(\rho) \right] \right| \leq \frac{1}{T} \sum_{t=0}^{T-1} \left| V_{r_p}^{\pi_t}(\rho) - \hat{V}_{r_p}^{\pi_t}(\rho) \right|$$

$$\leq \frac{1}{T} \sum_{t=0}^{T-1} \left\| V_{r_p}^{\pi_t} - \hat{V}_{r_p}^{\pi_t} \right\|_\infty$$

Recall that $\hat{M}_{r+\lambda_t c}$ satisfies the gap condition with $\iota = \frac{\omega\,\delta}{30\,|\Lambda||S||A|^2}$ for every $\lambda_t \in \Lambda$. Since $|\Lambda| = \frac{U}{\varepsilon_l}$, $\iota = \frac{\omega\,\delta(1-\gamma)\,\varepsilon_l}{30\,U|S||A|^2}$. Since $\pi_t := \arg\max \hat{V}_{r_p + \lambda_t c}^\pi$, we use Lemma 5 with $\alpha = r_p + \lambda_t c$ and $\beta = r_p$, and obtain the following result. For $N \geq \frac{4\,C(\delta)}{1-\gamma}$, for each $t \in [T]$, with probability at least $1 - \delta/5$,

$$\left\| V_{r_p}^{\pi_t} - \hat{V}_{r_p}^{\pi_t} \right\|_\infty \leq \sqrt{\frac{C(\delta)}{N \cdot (1-\gamma)^3}} (1+\omega) \leq 2\sqrt{\frac{C(\delta)}{N \cdot (1-\gamma)^3}}$$

Using the above relations,

$$\left| V_{r_p}^{\bar{\pi}_T}(\rho) - \hat{V}_{r_p}^{\bar{\pi}_T}(\rho) \right| \leq 2\sqrt{\frac{C(\delta)}{N \cdot (1-\gamma)^3}}$$

Similarly, invoking Lemma 5 with $\alpha = r_p + \lambda_t c$ and $\beta = c$ gives the bound on $\left| V_c^{\bar{\pi}_T}(\rho) - \hat{V}_c^{\bar{\pi}_T}(\rho) \right|$. $\square$

---

**Lemma 7.** *For $\delta \in (0,1)$, $\omega \leq 1$ and $C'(\delta) = 72 \log \left( \frac{4|S| \log(e/1-\gamma)}{\delta} \right)$, if $N \geq \frac{4\,C'(\delta)}{1-\gamma}$ and $B(\delta,N) := \sqrt{\frac{C'(\delta)}{(1-\gamma)^3 N}}$, then with probability at least $1 - 3\delta$,*

$$\left| V_{r_p}^{\pi^*}(\rho) - \hat{V}_{r_p}^{\pi^*}(\rho) \right| \leq 2B(\delta,N)\,; \; \left| V_c^{\pi^*}(\rho) - \hat{V}_c^{\pi^*}(\rho) \right| \leq B(\delta,N)\,; \; \left| V_c^{\pi_c^*}(\rho) - \hat{V}_c^{\pi_c^*}(\rho) \right| \leq B(\delta,N).$$

*Proof.* Since $\pi^*$ and $\pi_c^*$ are fixed policies independent of the sampling, we can directly use Li et al. [21, Lemma 1]. $\square$

### D.1  Helper lemmas

**Lemma 16.** *With probability at least $1 - \delta/10$, for every $\lambda \in \Lambda$, $\hat{M}_{r_p + \lambda c}$ satisfies the gap condition in Definition 4 with $\iota = \frac{\omega\,\delta\,(1-\gamma)}{30\,|\Lambda||S||A|^2}$.*

*Proof.* Using Lemma 18 with a union bound over $\Lambda$ gives the desired result. $\square$

**Lemma 17.** *Let $\pi_\alpha^*$ and $\pi_{\alpha'}^*$ be two optimal policies for an MDP with rewards $\alpha$ and $\alpha'$ respectively. Suppose $Q_\alpha^*$ satisfies the $\iota$-gap condition. Then, for all $\alpha'$ with $\|\alpha' - \alpha\|_\infty < \iota(1-\gamma)/2$, we have*

$$\pi_\alpha^* = \pi_{\alpha'}^*.$$

*Proof.* Since $\|\alpha' - \alpha\|_\infty < \iota(1-\gamma)/2$, we thus have

$$\|Q_\alpha^* - Q_{\alpha'}^*\|_\infty < \frac{\iota(1-\gamma)}{2(1-\gamma)} = \frac{\iota}{2}.$$

Note that, for all $s$,

$$\begin{aligned}
Q_\alpha^*(s, \pi_{\alpha'}^*(s)) &> Q_{\alpha'}^*(s, \pi_{\alpha'}^*(s)) - \frac{\iota}{2} \\
&\geq Q_{\alpha'}^*(s, \pi_\alpha^*(s)) - \frac{\iota}{2}, \\
&> Q_\alpha^*(s, \pi_\alpha^*(s)) - \iota \\
&\geq Q_\alpha^*(s, a'). \quad \forall a' \neq \pi_\alpha^*(s).
\end{aligned}$$

Hence, $\pi_{\alpha'}^*(s) \notin \{a' : a' \neq \pi_\alpha^*(s)\}$ for all $s \in S$, and consequently $\pi_\alpha^* = \pi_{\alpha'}^*$. $\qquad\square$

---

**Lemma 18** (Lemma 6 of Li et al. [21]). *Consider the MDP $M = (S, A, P, \gamma, r_p)$ with randomly perturbed rewards $r_p$ ($r_p(s, a) = r(s, a) + \xi(s, a)$ where $\xi(s, a) \sim \mathcal{U}[0, \omega]$ and $r(s, a) \in \mathbb{R}$). If optimal Q-function is denoted as $Q_{r_p}^*$ and $\pi_{r_p}^*$ is an optimal deterministic policy for $M$ then for any $\delta \in [0, 1]$, with probability at least $1 - \delta$ we have, for all $s$ and $a \neq \pi_{r'}^*(s)$*

$$Q_{r_p}^*(s, \pi_{r_p}^*(s)) \geq Q_{r_p}^*(s, a) + \frac{\omega\delta(1-\gamma)}{3|S||A|^2}$$

*i.e. $M$ satisfies the gap condition in Definition 4 with $\iota = \frac{\omega\delta(1-\gamma)}{3|S||A|^2}$.*

---

**Lemma 19** (Bernstein inequality). *Fix a state $s$, an action $a$. Let $\delta \geq 0$. Then, for any fixed vector $V$, with probability greater than $1 - \delta$, it holds that,*

$$\left|(\mathcal{P}_{s,a} - \hat{\mathcal{P}}_{s,a}) \cdot V\right| \leq \sqrt{\frac{2\log(4/\delta)}{N}}\sqrt{Var_{P_{s,a}}(V)} + \frac{2\log(4/\delta)\,\|V\|_\infty}{3N}.$$

# E  Lower Bound

In this section, we first present the lower-bound in the simplified bandit setting (Appendix E.1), and then present the formal CMDP lower bound in Appendix E.2.

## E.1  Bandit Setting

Consider the 2-arm bandit setting where arm-1 has a mean reward $H := \frac{1}{1-\gamma}$ and mean constraint reward $b - \zeta$ and arm-2 has a mean reward $0$ and mean constraint reward $b + \zeta$. For the constraint RHS equal to $b$ (implying that the Slater constant is $\zeta$), the ground truth optimal policy plays each arm with equal probability to achieve a reward value $H/2$ (with no constraint violation).

However, suppose there is an error $\varepsilon'$ in the estimation of the constraint reward in arm-2 (we estimate it to be $b + \zeta - \varepsilon'$); then even if everything else is exactly estimated, the empirical optimal policy has to play action 2 with probability $\zeta/(2\zeta - \varepsilon') \approx 1/2 + \varepsilon'/(4\zeta)$ to satisfy the constraint, which gives a value $H/2 - H\varepsilon'/(4\zeta)$. This results in an $H\varepsilon'/(4\zeta)$ error of the final policy. To obtain an $\varepsilon$-correct policy without constraint violation, one needs to set $\varepsilon' = \varepsilon\zeta/H = \varepsilon\zeta(1-\gamma)$, thus inflating the sample complexity compared to the unconstrained setting. We instantiate this intuition for CMDPs in Theorem 8.

## E.2  Proof of Theorem 8

**Theorem 8.** There exists constants $\gamma_0 \in (1 - 1/\log(|\mathcal{S}|), 1)$, $0 \leq \varepsilon_0 \leq \frac{1}{(1-\gamma)} \min\left\{1, \frac{\gamma}{(1-\gamma)\zeta}\right\}$, $\delta_0 \in (0,1)$, such that, for any $\gamma \in (\gamma_0, 1)$, $\varepsilon \in (0, \varepsilon_0)$, $\delta \in (0, \delta_0)$, any $(\varepsilon, \delta)$-sound algorithm requires $\Omega\left(\frac{SA\ln(1/4\delta)}{\varepsilon^2\zeta^2(1-\gamma)^5}\right)$ samples from the generative model in the worst case.

*Proof of Theorem 8.* Without loss of generality, we assume $|\mathcal{S}| = 2^m - 1$ for some integer $m$. In what follows, we first introduce our hard instance. Note that some of the states of this instance may have less than $|\mathcal{A}|$ actions. This is without loss of generality as one can easily duplicate actions to make each state having exactly $|\mathcal{A}|$ actions.

**Hard Instance.** We will consider the basic gadget defined in Figure 2. We will make $|\mathcal{S}| + 1$ copies of this gadget. In the $i$-th gadet for $i = 0, 1, \ldots, |\mathcal{S}|$, there is an input state $s_i$ with 2 actions $a_l, a_r$. Playing $a_l$ deterministically transitions to $\tilde{s}_i$ with reward and constraint 0. Playing $a_r$ goes to $z_i'$ with reward and constraint 0. In state $\tilde{s}_i$, there are $|\mathcal{A}|$ actions (only 1 action $a_0$ on $\tilde{s}_0$). Playing action $a \in \mathcal{A}$, this state transitions to state $z_i$ with probability $1 - p_{i,a}$ and self loop with probability $p_{i,a}$, where $p_{i,a}$ will be determined shortly. The reward at this state is $1$ and constraint reward is $u$ to be determined. In state $z_i'$, there is only one action, whose reward is 0 and constraint is $(b + \zeta)(1 - \gamma)/\gamma^{m+3}$.

Lastly, we consider $2|\mathcal{S}| + 1$ routing state $o_0, o_1, \ldots$ that form a binary tree, i.e., in each of these routing state, there are two actions $a_0$ and $a_1$. The state-action pair $(o_i, a_j)$ transitions to state $o_{2i+j+1}$ for $i < |\mathcal{S}|$. Each state $o_{|\mathcal{S}|+k}$ transitions deterministically to the gate state $s_k$ for $k = 0, 1, \ldots |\mathcal{S}|$. Rewards and constraint are all 0 for these states. Note that, for any state $s_k$, there is a unique deterministic path of length $m + 1$ connecting $o_0$. Hence we require $\gamma \geq 1 - 1/(cm)$ for some constant $c$ and hence $\gamma^{\Theta(m)} = \Theta(1)$.

Note that this instance modifies the MDP instance in [14]. Some parameters chosen are also adapted from there.

**Hypotheses.** With the above defined hard instance structure, we now define a family of hypotheses on the probability transitions. Later we will show that any sound algorithm would be able to test the hypothesises but would require a large number of samples. Let $q_0, q_1, q_2 \in (0, 1)$ be some parameters to be determined. We define,

- *Null case* MDP $M_0$: $p_{0,a_0} = q_1$, and $p_{i,a} = q_0$ for all $i \in [|\mathcal{S}|]$ and $a \in \mathcal{A}$.

- *Alternative case* MDP $M_{i,a}$: $p_{0,a_0} = q_1$, $p_{j,a'} = q_0$ for all $(j, a') \neq (i, a)$, and $p_{i,a} = q_2$.

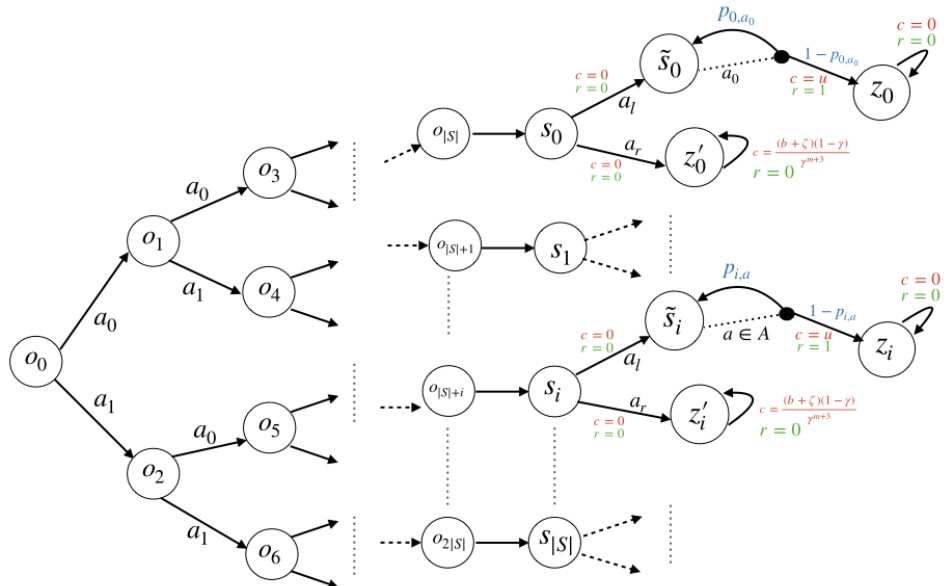

Figure 2: The lower-bound instance consists of CMDPs with $S = 2^m - 1$ (for some integer $m > 0$) states and $A$ actions. We consider $SA + 1$ CMDPs – $M_0$ and $M_{i,a}$ ($i \in \{1, \ldots S\}$, $a \in \{1, \ldots, A\}$) that share the same structure shown in the figure. For each CMDP, $o_0$ is the fixed starting state and there is a deterministic path of length $m + 1$ from $o_0$ to each of the $S + 1$ states – $s_i$ (for $i \in \{0, 1, \ldots, S\}$). Except for states $\tilde{s}_i$, the transitions in all other states are deterministic. For $i \neq 0$, for action $a \in \mathcal{A}$ in state $\tilde{s}_i$, the probability of staying in $\tilde{s}_i$ is $p_{i,a}$, while that of transitioning to state $z_i$ is $1 - p_{i,a}$. There is only one action $a_0$ in $\tilde{s}_0$ and the probability of staying in $\tilde{s}_0$ is $p_{0,a_0}$, while that of transitioning to state $z_0$ is $1 - p_{0,a_0}$. The CMDPs $M_0$ and $M_{i,a}$ only differ in the values of $p_{i,a}$. The rewards $r$ and constraint rewards $c$ are the same in all CMDPs and are denoted in green and red respectively for each state and action.

Note that all these MDPs $M_0, M_1 \ldots$ share exactly the same graph structure, with only the transition probability changes. Moreover, $M_{i,a}$ differs from $M_0$ only on state-action pair $(\tilde{s}_i, a)$.

**Optimal Policies.** Now we specify $q_0, q_1$ and $q_2$ and check the optimal policies of each hypothesis. Let $\varepsilon' = (1 - \gamma)\zeta\varepsilon$,

$$q_0 = \frac{1 - c_1(1 - \gamma)}{\gamma}, q_1 = q_0 + \alpha_1, \text{ and } q_2 = q_0 + \alpha_2,$$

and

$$\alpha_1 = \frac{c_2(1 - \gamma q_0)^2 \varepsilon'}{\gamma}, \quad \text{and} \quad \alpha_2 = \frac{c_3(1 - \gamma q_0)^2 \varepsilon'}{\gamma}$$

for some absolute constants $c_1, c_2, c_3 > 0$ and that $\alpha_1/q_0 \in (0, 1/2)$, $\alpha_1/(1 - q_0) \in (0, 1/2)$, $\alpha_2/q_0 \in (0, 1/2)$, and $\alpha_2/(1 - q_0) \in (0, 1/2)$. We choose these parameters such that

$$\frac{1}{c_1(1 - \gamma)} = \frac{1}{1 - \gamma q_0} < \frac{1}{1 - \gamma q_1} < \frac{1}{1 - \gamma q_2} = \frac{1}{c_1(1 - \gamma)} + c_4 \varepsilon'$$

for some constants $c_4 > 0$ and that

$$\left| \frac{1}{1 - \gamma q_1} - \frac{1}{1 - \gamma q_0} \right| = \Theta(\varepsilon'), \text{ and } \left| \frac{1}{1 - \gamma q_1} - \frac{1}{1 - \gamma q_2} \right| = \Theta(\varepsilon').$$

Note that, for the reward values, if $\zeta = \Theta(1)$, then these actions only differ by $\Theta(\varepsilon') \ll \varepsilon$. A correct algorithm would not need to distinguish these actions. Yet, once the constraints are concerns, we will show shortly that these actions do differ because of the constraint values.

With these parameters, we can derive the optimal CMDP policy for $M_0$ and each $M_{i,a}$. For any policy $\pi$, we denote the state occupancy measure as $\mu^\pi$, i.e.,

$$\mu^\pi(s, a) = \sum_z \rho(z) \sum_{t=1}^\infty \gamma^{t-1} \sum_z \Pr_\pi[s_t = s, a_t = a | s_1 = z] \tag{22}$$

where $\rho$ is the initial distribution with $\rho(o_0) = 1$ and $\rho(s) = 0$ for all $s \neq o_0$. This occupancy measure describes the discounted reachablity from $o_0$ to an arbitrary state action pair. The reward value and constraint value can be written as

$$V^\pi = \sum_{s,a} \mu^\pi(s, a) r(s, a), \quad \text{and} \quad V_c^\pi = \sum_{s,a} \mu^\pi(s, a) c(s, a).$$

Note that, given a state-occupancy measure $\mu$, a policy can be specified as $\pi_\mu(a|s) = \mu(s, a) / \sum_{a'} \mu(s, a')$. We can use the LP formulation for the CMDP as follows

$$\max \sum_{s,a} \mu(s, a) r(s, a) \text{ subject to },$$

$$\forall s: \quad \sum_a \mu(s, a) = \rho(s) + \gamma \sum_{s',a'} P(s|s', a') \mu(s', a'),$$

$$\sum_{s,a} \mu^\pi(s, a) c(s, a) \geq b,$$

$$\mu(s, a) \geq 0. \tag{23}$$

Due to the structure of the CMDPs, we can further simply the structure of the constraints. In particular, we have

$$\forall i: \quad \sum_a \mu(\tilde{s}_i, a) = \gamma \mu(s_i, a_l) + \gamma \sum_a p_{i,a} \mu(\tilde{s}_i, a) \tag{24}$$

let $\bar{\nu} = \sum_i \mu(s_i, a_r)$. We then have

$$\sum_i \mu(s_i, a_l) + \bar{\nu} = \gamma^{m+2},$$

and

$$\mu(z_i', a) = \frac{\gamma \mu(s_i, a_r)}{1 - \gamma}.$$

Consider $M_0$, let $\mu_0 := \mu(s_0, a_l)$, $\mu_0^c := \sum_{i>0} \mu(s_i, a_l)$ we have

$$\sum_{i>0,a} \mu(\tilde{s}_i, a) = \frac{\gamma \mu_0^c}{1 - \gamma q_0}, \quad \text{and} \quad \mu(\tilde{s}_0, a_0) = \frac{\gamma \mu_0}{1 - \gamma q_1}.$$

The LP can be rewritten as

$$\max \quad \frac{\gamma \mu_0}{1 - \gamma q_1} + \frac{\gamma \mu_0^c}{1 - \gamma q_0}$$

$$\text{s.t.} \frac{\gamma \mu_0 \cdot u}{1 - \gamma q_1} + \frac{\gamma \mu_0^c \cdot u}{1 - \gamma q_0} + \frac{\bar{\nu} \cdot (b + \zeta)}{\gamma^{m+2}} \geq b, \ \mu_0 + \mu_0^c + \bar{\nu} = \gamma^{m+2}, \text{ and } \mu(s, a) \geq 0, \ \forall(s, a).$$

Here, we specify the values of $u$ as,

$$u = \frac{(1 - \gamma q_0)(b - x)}{\gamma^{m+3}},$$

for some $x = \Theta(\zeta)$, such that,

$$\frac{\gamma^{m+3} u}{1 - \gamma q_0} = b - x < \frac{\gamma^{m+3} u}{1 - \gamma q_1} = b - x + \varepsilon_1' < \frac{\gamma^{m+3} u}{1 - \gamma q_2} = b - x + \varepsilon_2' < b,$$

where $c'\varepsilon' \leq \varepsilon_1' \leq \varepsilon_1' + c''\varepsilon' \leq \varepsilon_2' \leq c'''\varepsilon'$ for some positive constants $c', c'', c'''$ determined by $c_1, c_2, c_3$. For this value of $u$, the maximum value of the constraint value function $\max V_c^\pi$ is $b + \zeta$, implying that $\zeta$ is the Slater constant for all these CMDPs.

Thus, for $M_0$, the solution is,

$$\mu_0 = \frac{\gamma^{m+2}\zeta}{\zeta + x - \varepsilon_1'}, \quad \mu_0^c = 0, \quad \text{and} \quad \bar{\nu} = \frac{(x - \varepsilon_1')\gamma^{m+2}}{\zeta + x - \varepsilon_1'}.$$

Note that this implies the policy deterministically choose a path to reach state $s_0$, and then plays action $a_l$ with probability $\mu_0/\gamma^{m+2}$. The optimal value in this case is

$$V_{M_0}^*(o_0) = \frac{\zeta}{\zeta + x - \varepsilon_1'} \cdot \frac{\gamma^{m+3}}{1 - \gamma q_1}.$$

Similarly, for $M_{i,a}$ with $i \geq 1$, let $\mu_{i,a} := \mu(s_i, a)$, $\mu_{i,a}^c := \sum_{i'>0,(i',a')\neq(i,a)} \mu(s_{i'}, a')$, the LP can be written as

$$\max \quad \frac{\gamma\mu_0}{1 - \gamma q_1} + \frac{\gamma\mu_{i,a}^c}{1 - \gamma q_0} + \frac{\gamma\mu_{i,a}}{1 - \gamma q_2}$$

$$\text{s.t.} \quad \frac{\gamma\mu_0 u}{1 - \gamma q_1} + \frac{\gamma\mu_{i,a}^c u}{1 - \gamma q_0} + \frac{\gamma\mu_{i,a} u}{1 - \gamma q_2} + \frac{\bar{\nu}\cdot(b+\zeta)}{\gamma^{m+2}} \geq b, \ \mu_0 + \mu_{i,a}^c + \mu_{i,a} + \bar{\nu} = \gamma^{m+2},$$

$$\text{and } \mu(s,a) \geq 0, \ \forall(s,a).$$

the solution is

$$\mu_{i,a} = \frac{\zeta\gamma^{m+2}}{\zeta + x - \varepsilon_2'}, \quad \mu_{i,a}^c = \mu_0 = 0, \quad \text{and} \quad \bar{\nu} = \frac{(x - \varepsilon_2')\gamma^{m+2}}{\zeta + x - \varepsilon_2'},$$

i.e., the policy chooses a path to reach state $s_i$ deterministically and with optimal value

$$V_{M_{i,a}}^*(o_0) = \frac{\zeta}{\zeta + x - \varepsilon_2'} \cdot \frac{\gamma^{m+3}}{1 - \gamma q_2}.$$

Lastly, we shall check the gap of the value functions.

$$\left| V_{M_{i,a}}^*(o_0) - V_{M_0}^*(o_0) \right| \geq \left( \frac{\zeta}{\zeta + x - \varepsilon_2'} - \frac{\zeta}{\zeta + x - \varepsilon_1'} \right) \cdot \frac{\gamma^{m+3}}{1 - \gamma q_1} = \frac{\varepsilon''}{\zeta + x} \cdot \frac{\gamma^{m+3}}{1 - \gamma q_1} \geq c_7 \varepsilon.$$

where $\varepsilon'' \geq c_8 \varepsilon'$ for some constants $c_7, c_8$ determined by $c_1, c_2, c_3$ and $x$. Thus the error in $V_c^\pi$ is amplified by a factor of $\Theta[(1 - \gamma)^{-1}\zeta^{-1}]$.

**Implications of Soundness: Near-Optimal Policies.** Let $\mathcal{K}$ be a $(\varepsilon, \delta)$-sound algorithm, i.e., on input any CMDP with a generative model, it outputs a policy, which is $\varepsilon$-optimal with probability at least $1 - \delta$. We thus define the event

$$\mathcal{E}_0 = \left\{ \mathcal{K} \text{ outputs policy } \pi \text{ such that } \mu^\pi(s_0, a_l) \geq \frac{\zeta\gamma^{m+2}}{(\zeta + x - \varepsilon_1'/2)} \right\},$$

i.e., this event requires the output policy reaching $s_0$ and play action $a_l$ with sufficiently high probability. We now measure the probability of $\mathcal{E}_0$ on different input CMDPs. Due to the soundness, it must be the case that

$$\Pr_{M_{i,a}}[\mathcal{E}_0] < \delta.$$

If not, on $\mathcal{E}_0$, $\mu^\pi(s_0, a_l) \geq \frac{\zeta\gamma^{m+2}}{\zeta + x - \varepsilon_1'/2}$, we can then compute the best possible $V_{M_{i,a}}^\pi(o_0)$ as solving the following LP,

$$\max \quad \frac{\gamma\mu_0}{1 - \gamma q_1} + \frac{\gamma\mu_{i,a}^c}{1 - \gamma q_0} + \frac{\gamma\mu_{i,a}}{1 - \gamma q_2}$$

$$\text{s.t.} \quad \frac{\gamma\mu_0 u}{1 - \gamma q_1} + \frac{\gamma\mu_{i,a}^c u}{1 - \gamma q_0} + \frac{\gamma\mu_{i,a} u}{1 - \gamma q_2} + \frac{\bar{\nu}\cdot(b+\zeta)}{\gamma^{m+2}} \geq b, \ \mu_0 + \mu_{i,a}^c + \mu_{i,a} + \bar{\nu} = \gamma^{m+2},$$

$$\mu_0 \geq \frac{\zeta\gamma^{m+2}}{(\zeta + x - \varepsilon_1'/2)} \text{ and } \mu(s,a) \geq 0, \ \forall(s,a).$$

Plugging the values of $p_{i,a}$, and due to $q_0 < q_1 < q_2$, we obtain $\mu_0 = \frac{\zeta\gamma^{m+2}}{\zeta+x-\varepsilon_1'/2}$, $\mu_{i,a}^c = 0$ and,

$$\mu_{i,a} \cdot (b - x + \varepsilon_2') + \mu_0 \cdot (b - x + \varepsilon_1') + \bar{\nu}(b + \zeta) = b\gamma^{m+2}, \quad \mu_{i,a} + \mu_0 + \bar{\nu} = \gamma^{m+2}.$$

Hence,

$$\mu_{i,a} = \gamma^{m+2} \cdot \frac{\zeta - \mu_0(\zeta + x - \varepsilon_1')}{\zeta + x - \varepsilon_2'} \leq \frac{c_9\varepsilon_1'\zeta \cdot \gamma^{m+2}}{2(\zeta + x - \varepsilon_1'/2)(\zeta + x - \varepsilon_2)}$$

for some constant $c_9$ depending on $c_1, c_2, c_3, x$, and

$$V^\pi(o_0) \leq \frac{\zeta}{\zeta + x - \varepsilon_1'/2} \cdot \frac{\gamma^{m+3}}{1 - \gamma q_1} + \frac{c_9\varepsilon_1'\zeta}{2(\zeta + x - \varepsilon_1'/2)(\zeta + x - \varepsilon_2)} \cdot \frac{\gamma^{m+3}}{1 - \gamma q_2}$$

Note that

$$V_{M_{i,a}}^*(o_0) = \frac{\zeta}{\zeta + x - \varepsilon_2'} \cdot \frac{\gamma^{m+3}}{1 - \gamma q_2}$$

and

$$V_{M_{i,a}}^*(o_0) - V_{M_{i,a}}^\pi(o_0) \geq \left( \frac{\zeta}{\zeta + x - \varepsilon_2'} - \frac{\zeta}{\zeta + x - \varepsilon_1'/2} - \frac{c_9\varepsilon_1'\zeta}{2(\zeta + x - \varepsilon_1'/2)(\zeta + x - \varepsilon_2')} \right) \cdot \frac{\gamma^{m+3}}{1 - \gamma q_2} \geq \varepsilon$$

for some appropriately chosen $c_1, c_2, c_3, x$, which is a contradiction of the $(\varepsilon, \delta)$-soundness.

**Implications of Soundness: Expectation on Null Hypothesis.** Let $N_{i,a}$ be the number of samples the algorithm $\mathcal{K}$ takes on state-action $(s_i, a)$. Next we show that, $\mathbb{E}[N_{i,a}]$ has to be big on $M_0$.

---

**Lemma 20.** *Let $t_* = \frac{c_{10}\log\delta^{-1}}{(1-\gamma)^3\varepsilon'^2}$ for some constant $c_{10}$. For any $(\varepsilon, \delta)$-sound algorithm $\mathcal{K}$, for any $(i, a)$, we have*

$$\mathbb{E}_{M_0}(N_{i,a}) \geq t_*.$$

---

*Proof.* Suppose $\mathbb{E}_{M_0}(N_{i,a}) < t_*$, then we aim to show a contradiction: $\Pr_{M_{i,a}}[\mathcal{E}_0] \geq \delta$. Similar to the proof above, since $\mathcal{K}$ is $(\varepsilon, \delta)$-sound, it must be the case that

$$\Pr_{M_0}[\mathcal{E}_0] \geq 1 - \delta.$$

We now consider the likelihood ratio

$$\Pr_{M_{i,a}}[\mathcal{E}_0] / \Pr_{M_0}[\mathcal{E}_0].$$

For any realization of the empirical samples, consider the samples the algorithm takes as $\tau = \{(s_{i,a}^{(1)}, s_{i,a}^{(2)}, \ldots, s_{i,a}^{(N_{i,a})}) : (i, a) \in [|\mathcal{S}|] \times [\mathcal{A}]\}$. Let us define $N_{s',s,a}$ as the number of samples from $(s, a) \to s'$. By Markov property, since the only difference of the probability matrix between $M_{i,a}$ and $M_0$ is on $p_{i,a}$, we have

$$\frac{\Pr_{M_{i,a}}[\tau]}{\Pr_{M_0}[\tau]} = \frac{q_2^{N_{\tilde{s}_i,\tilde{s}_i,a}}(1 - q_2)^{N_{z_i,\tilde{s}_i,a}}}{q_0^{N_{\tilde{s}_i,\tilde{s}_i,a}}(1 - q_0)^{N_{z_i,\tilde{s}_i,a}}} = \left(\frac{q_2}{q_0}\right)^{N_{\tilde{s}_i,\tilde{s}_i,a}} \cdot \left(\frac{1 - q_2}{1 - q_0}\right)^{N_{z_i,\tilde{s}_i,a}}$$

$$= \left(1 + \frac{\alpha_2}{q_0}\right)^{Nq_0 - \Delta} \cdot \left(1 - \frac{\alpha_2}{1 - q_0}\right)^{N(1-q_0) + \Delta}$$

where $\Pr_M[\tau]$ denotes the probability of $\mathcal{K}$ taking the samples $\tau$ in CMDP $M$, $\Delta = N_{i,a}q_0 - N_{\tilde{s}_i,\tilde{s}_i,a}$, and $N = N_{i,a}$. By a similar derivation of Lemma 5 in [14] (page 15-19), on the following event,

$$\mathcal{E}_{i,a}' = \left\{ N_{i,a} \leq 10t_*, \text{ and } |N_{\tilde{s}_i,\tilde{s}_i,a} - N_{i,a}q_0| \leq \sqrt{20(1 - q_0)q_0N_{i,a}} \right\}$$

we have

$$\frac{\Pr_{M_{i,a}}[\tau]}{\Pr_{M_0}[\tau]} \geq 4\delta$$

provided appropriately chosen $c_1, c_2, c_3, x, c_{10}$. By Markov inequality and Doob's inequality (e.g. Lemma 7-8 of [14]), we have

$$\Pr_{M_0}[\mathcal{E}'_{i,a}] \geq 1 - \frac{1}{10} - \frac{1}{10} = \frac{4}{5}.$$

We are able to compute the probability of $\mathcal{E}_0$ on $M_{i,a}$ as follows:

$$\Pr_{M_{i,a}}[\mathcal{E}_0] = \sum_{\tau \in \mathcal{E}_0} \Pr_{M_{i,a}}[\tau] \geq \sum_{\tau \in \mathcal{E}_0 \cap \mathcal{E}'_{i,a}} \Pr_{M_{i,a}}[\tau]$$

$$= \sum_{\tau \in \mathcal{E}_0 \cap \mathcal{E}'_{i,a}} \frac{\Pr_{M_{i,a}}[\tau]}{\Pr_{M_0}[\tau]} \cdot \Pr_{M_0}[\tau] \geq 4\delta \sum_{\tau \in \mathcal{E}_0 \cap \mathcal{E}'_{i,a}} \Pr_{M_0}[\tau] \geq 3\delta,$$

provided $\delta \leq c_{11}$ for some absolute constant $c_{11}$, hence a contradiction of soundness. $\square$

**Wrapping up.**   Hence, if the algorithm is $(\varepsilon, \delta)$-sound for all $\{M_{i,a}\}$, it must be the case that

$$\mathbb{E}_{M_0}[N_{i,a}] \geq t_*, \forall (i,a) \in [|\mathcal{S}|] \times \mathcal{A}.$$

By linearity of expectation, we have

$$\mathbb{E}_{M_0}\left[\sum_{i,a} N_{i,a}\right] \geq |\mathcal{S}||\mathcal{A}|t_*.$$

Since $\varepsilon' = \varepsilon(1-\gamma)\zeta$, $\mathbb{E}_{M_0}\left[\sum_{i,a} N_{i,a}\right] \geq \frac{c_{10}|\mathcal{S}||\mathcal{A}| \log \delta^{-1}}{(1-\gamma)^5 \zeta^2 \varepsilon^2}$, which completes the proof. $\square$

# F  Estimating $\zeta$

In this section, we show that $\zeta$ can be estimated up to error $0.2\zeta$ using small number of samples. The formal guarantee is provided in the following theorem.

**Theorem 21.** There exists an algorithm that, with probability at least $1 - \delta$, halts, takes

$$O\left(\frac{c_{max}\log\left(\frac{|\mathcal{S}||\mathcal{A}|}{(1-\gamma)\delta\zeta}\right)}{(1-\gamma)^3\zeta^2}\right)$$

samples per state-action pair, and outputs an estimator $\hat{\zeta}$, such that

$$|\hat{\zeta} - \zeta| \leq 0.2\zeta.$$

*Proof.* Let $\zeta_i = 2^{-i}/(1-\gamma)$ for $i = 0, 1, 2, \ldots$, and

$$N_i = \frac{c_{\max}C_i(\delta)}{(1-\gamma)^3\zeta_i^2},$$

where

$$C_i(\delta) = c'\log\left(2\frac{|\mathcal{S}||\mathcal{A}|i^2}{(1-\gamma)\zeta_i\delta}\right)$$

for some constant $c'$. We start by running the algorithm in [21] for $N_i$ samples per state-action pair on the MDP with $c$ as reward , for $i = 0, 1, \ldots$ and stop if the following is satisfied:

$$|\hat{V}_{c,i}^*(\rho) - b| \geq 9\zeta_i,$$

where $\hat{V}_{c,i}^*$ is the empirical optimal value function obtained for using $N_i$ samples. Then we output $\hat{\zeta} = \hat{V}_{c,i}^*(\rho)$.

Next we show that the algorithm halts. Let $\mathcal{E}_i$ be event for iteration $i$,

$$|\hat{V}_{c,i}^*(\rho) - V_c^*(\rho)| \leq \zeta_i.$$

Thus, by Theorem 1 of [21],

$$\Pr[\mathcal{E}_i] \geq 1 - \frac{\delta}{2i^2}.$$

Next, let $i^*$ be such that $0.05\zeta \leq \zeta_{i^*} < 0.1\zeta$. Hence, if $\mathcal{E}_{i^*}$ happens, then

$$|\hat{V}_{c,i^*}^*(\rho) - V_c^*(\rho)| \leq \zeta_{i^*}$$

and

$$|V_c^*(\rho) - b| - |\hat{V}_{c,i^*}^*(\rho) - V_c^*(\rho)| \leq |\hat{V}_{c,i^*}^*(\rho) - b|.$$

Hence, on $\mathcal{E}_{i^*}$,

$$|\hat{V}_{c,i^*}^*(\rho) - b| \geq \zeta - \zeta_{i^*} \geq 0.9\zeta \geq 9\zeta_{i^*}$$

and the algorithm halts at least before iteration $i^*$.

Next, suppose the algorithm halts at $i \leq i^*$, then on $\mathcal{E}_i$, we have

$$|\hat{V}_{c,i}^*(\rho) - V_c^*(\rho)| \leq \zeta_i,$$

$$|\hat{V}_{c,i}^*(\rho) - b| \geq 9\zeta_i \geq 9|\hat{V}_{c,i}^*(\rho) - V_c^*(\rho)|,$$

and

$$|(|\hat{V}_{c,i}^*(\rho) - b| - |V_{c,i}^*(\rho) - b|)| \leq |\hat{V}_{c,i}^*(\rho) - V_c^*(\rho)| \leq |\hat{V}_{c,i}^*(\rho) - b|/9.$$

Note that $\zeta = |V_c^*(\rho) - b|$, we have

$$|\hat{\zeta} - \zeta| \leq \hat{\zeta}/9 \implies \zeta \geq 8/9\hat{\zeta} \text{ and } |\hat{\zeta} - \zeta| \leq \zeta/8.$$

Thus, on event $\mathcal{E} = \mathcal{E}_1 \cap \mathcal{E}_2 \cap \mathcal{E}_3 \ldots \mathcal{E}_{i^*}$, which happens with probability at least

$$1 - \sum_{i=1}^{i^*} \frac{\delta}{2i^2} \geq 1 - \frac{\pi^2 \delta}{12},$$

we have $|\hat{\zeta} - \zeta| \leq \zeta/8$, proving the correctness.

We now consider the overall sample complexity. Suppose $\mathcal{E}$ happens, then the number of samples consumed is upper bounded by

$$\sum_{i=1}^{i^*} N_i \leq \frac{c'' c_{max} C_{i^*}(\delta)}{(1-\gamma)^3 \zeta^2} = \frac{c'' c' c_{max} \log\left(\frac{|\mathcal{S}||\mathcal{A}|}{(1-\gamma)\delta\zeta}\right)}{(1-\gamma)^3 \zeta^2},$$

for some constant $c''$, completing the proof. □

# G   Comparison to Bai et al. [6]

For a target error of $\varepsilon$, our lower-bound construction in the strict feasibility setting (Section 7) shows that it is important to estimate the constraint value function to a smaller error equal to $\varepsilon'$. Intuitively, this is because a small estimation error in the constraint value can (incorrectly) render the optimal policy infeasible and result in a $\frac{\text{Range(value function)}}{\zeta}$ suboptimality gap.

For the theoretical results in [6], the value functions are normalized and hence Range(value function) $= 1$. Hence, a small constraint violation can result in an $\frac{1}{\zeta}$ suboptimality gap, and the constraint value function needs to be estimated to a smaller error equal to $\varepsilon' := \varepsilon\zeta$. Combining this with the standard results for the primal-dual algorithm for unconstrained MDPs [32] for *normalized value functions*, this implies a sample-complexity of $O\left(\frac{SA}{(1-\gamma)^2\,\varepsilon'^2}\right) = O\left(\frac{SA}{(1-\gamma)^2\,\varepsilon^2\zeta^2}\right)$ which is the sample complexity reported in [6, Theorem 2].

On the other hand, if we scale the value function to lie to the standard $O\left(1/1-\gamma\right)$ range, a small constraint violation can result in an $\frac{1}{(1-\gamma)\zeta}$ suboptimality gap, and the constraint value function needs to be estimated to a smaller error equal to $\varepsilon' := \varepsilon(1-\gamma)\zeta$. The rescaling also affects the sample complexity for the primal-dual algorithm for unconstrained MDPs [32]. Specifically, for unconstrained MDPs, if the value functions lie in the $[0, 1/1-\gamma]$ range, the primal-dual algorithm in Wang [32] requires $O\left(\frac{SA}{(1-\gamma)^4\,\varepsilon^2}\right)$ samples. Since we require a smaller error $\varepsilon'$ in the strict feasibility setting, this implies an $O\left(\frac{SA}{(1-\gamma)^4\,\varepsilon'^2}\right) = O\left(\frac{SA}{(1-\gamma)^6\,\varepsilon^2\zeta^2}\right)$ sample complexity.