# OpenReview forum: "Near-Optimal Sample Complexity Bounds for Constrained MDPs"
_NeurIPS.cc/2022/Conference — NeurIPS 2022 Accept_

### Official Review · Reviewer_KeTv · 2022-07-09

**Rating:** 6
**Confidence:** 4
**Soundness:** 3 good
**Presentation:** 2 fair
**Contribution:** 4 excellent

**Summary:**

This paper studies sample complexity bounds for constraint MDPs with generative model under relaxed feasibility setting (where the algorithm needs to find a policy that achieves \epsilon-optimal reward and \epsilon constraint violation) and strict feasibility setting (where the algorithm needs to find a policy that achieves \epsilon-optimal reward and 0 constraint violation). For the relaxed feasibility setting, this paper achieves $O(SA(1-\gamma)^{-3}\epsilon^{-2})$ sample complexity, which is minimax optimal and matches the lower bound for unconstraint case. For strict feasibility setting, this paper achieves $O(SA(1-\gamma)^{-5}\epsilon^{-2}\zeta^{-2})$ sample complexity and proves a corresponding lower bound.


**Questions:**

- The dependency on Slater constant ($\zeta=\max_\pi V^\pi_c(\rho)-b$) is interesting. Since the Slater constant doesn’t depend on the reward function, for any MDP one can always add a redundant state with r=0 and c=1 (and an action from the initial state that deterministically leads to this redundant state) so that the Slater constant becomes bigger. Consequently, the MDP becomes easier. If I understand correctly, with this redundant state the algorithm can somewhat “amortize” its reward and constraint: if the algorithm visits this redundant state with positive probability, its constraint will be increased but reward will be decreased. While the strict feasibility setting appears to be important in practice (e.g., we don’t want to break a robot arm), the behavior of the algorithm is somewhat undesired (it may output a mixture of an unsafe policy and extra-safe policy). Is there a way to address this mismatch?

- Do the bounds in [1] or [2] translated to the relaxed or strict feasibility setting? How does the bounds compare with each other?


**Limitations:**

The authors adequately addressed the limitations and potential negative societal impact.

**Strengths And Weaknesses:**

Originality. Although similar primal-dual methods are also considered in previous works, this paper provides novel analysis and therefore proves stronger results. The lower bound for the strict feasibility case is interesting and novel.

The following two related works are missing. In particular, Section 6 of [1] proves $O(\sqrt{SAT})$ regret for constrained MDPs without generative models, and [2] proves sample complexity bounds.

[1] Yu, Tiancheng, et al. Provably efficient algorithms for multi-objective competitive rl

[2] Miryoosefi, Sobhan, and Chi Jin. A simple reward-free approach to constrained reinforcement learning.


Quality. This paper is technically sound. I didn’t read through the appendix but the proofs that I read are morally correct.

Clarity. The exposition of the results could be improved:
-	The proof of Theorem 8 looks like a zero-knowledge proof and is hard to follow. The choice of constants are not explicitly stated, and some of the core ideas are hidden in the proof details. The intuition of why strict feasibility setting is harder than relaxed feasibility setting is unclear to me, as well as the idea of the hard instances.
-	While this paper proves a separation between relaxed and strict feasibility cases, the proof sketch for Theorem 2 & 3 looks very similar except the choice of parameters (e.g., b’, \epsilon_1). However, the choice of parameters is not justified in the first 9 pages. In particular, it’s unclear why the parameters have to be different without reading the full proof.

Significance. This paper proves minimax optimal sample complexity for constrained MDPs with generative model, and improves upon previous results. This paper partially answers the question that whether constrained MDPs are harder than unconstrained MDPs --- somewhat surprisingly, constrained MDPs with generatively model are not harder with relaxed feasibility.
For the strict feasibility case, the error decomposition (lemma 12) is novel and could be a useful tool to analyze constrained MDPs in different settings. The matching lower bound in this case demonstrates that the strict feasibility setting is strictly harder than the relaxed feasibility setting, which is an interesting result by itself.

---

> ### Author Response · Authors · 2022-08-02
> **Response to Reviewer KeTv**
>
> We thank the reviewer for their feedback and positive review of our work. We will add more intuition for our technical results and improve the paper's exposition. We answer their main questions below.
>
> *The following two related works are missing. In particular, Section 6 of [1] proves regret for constrained MDPs without generative models, and [2] proves sample complexity bounds. [1] Yu, Tiancheng, et al. Provably efficient algorithms for multi-objective competitive rl. [2] Miryoosefi, Sobhan, and Chi Jin. A simple reward-free approach to constrained reinforcement learning. Do the bounds in [1] or [2] translated to the relaxed or strict feasibility setting? How do the bounds compare with each other?*
>
> Thank you for providing these additional references. We will cite them in the final version of the paper. Both [1] and [2] are concerned with regret minimization in the relaxed feasibility setting assuming a finite horizon. While [1] considers the setting where the rewards are known, [2] considers the reward-free setting. Since both these works are concerned with the episodic finite-horizon setting (and their algorithms heavily make use of this structure), it is unclear how to extend their results to the infinite-horizon discounted setting studied in our paper.
>
> *The exposition of the results could be improved: - The proof of Theorem 8 looks like a zero-knowledge proof and is hard to follow. The choice of constants are not explicitly stated, and some of the core ideas are hidden in the proof details. The intuition of why a strict feasibility setting is harder than relaxed feasibility setting is unclear to me, as well as the idea of the hard instances. - While this paper proves a separation between relaxed and strict feasibility cases, the proof sketch for Theorem 2, 3 looks very similar except for the choice of parameters (e.g., $b’, \epsilon_1$). However, the choice of parameters is not justified in the first 9 pages. In particular, it’s unclear why the parameters have to be different without reading the full proof.*
>
> For a more complete description of the lower-bound construction (Theorem 8), please refer to Figure 2 in Appendix E. In the final version of the paper, we will move this figure and the corresponding description to the main paper.
>
> We now give some intuition about why there is a separation between the relaxed and strict feasibility settings. For the relaxed feasibility case, we can use a $b'$ (RHS of the constraint in the empirical CMDP) $\leq b$. Intuitively, the (small) estimation error in $\mathcal{P}$ translates into the (small) allowed constraint violation. On the other hand, for the strict feasibility setting, since even small constraint violations are not allowed, intuitively, we need to use a tighter (stricter) constraint for the empirical CMDP to account for the estimation error in $\mathcal{P}$. This requires using a $b' > b$. This changes the regret decomposition for the two settings (Lemma 11 vs Lemma 12) and introduces the additional sensitivity error term for the strict feasibility setting. The sensitivity error in the regret decomposition measures the change in the optimal reward value for the empirical CMDP when the constraint RHS is changed from $b$ to $b'$. This additional term is responsible for the increased sample complexity in the strict feasibility setting.
>
> We will include such an intuitive explanation in the final version of the paper. For further intuition about the difficulty in the strict feasibility setting, please also refer to the simplified lower bound construction in our response to Reviewer KfEz.
>
> *While the strict feasibility setting appears to be important in practice (e.g., we don’t want to break a robot arm), the behavior of the algorithm is somewhat undesired (it may output a mixture of an unsafe policy and extra-safe policy). Is there a way to address this mismatch?*
>
> Thank you for the good question -- at this moment, we can only bound the constraint violation and reward suboptimality for the mixture policy $\bar{\pi}_T$. This is similar to most results in the literature (e.g. [6,12] in our references). In order to output a policy that always satisfies the constraint i.e. is always safe, we require additional assumptions (for example in ``Amani et al, Safe Reinforcement Learning with Linear Function Approximation''). We do agree that this is an important research direction to consider in the future.

---

> > ### Comment · Reviewer_KeTv · 2022-08-08
> > **Thanks for the response!**
> >
> > I thank the authors for the detailed response! My concerns have been addressed.

---

### Official Review · Reviewer_KfEz · 2022-07-10

**Rating:** 5
**Confidence:** 5
**Soundness:** 2 fair
**Presentation:** 3 good
**Contribution:** 2 fair

**Summary:**

The paper considers Constrained MDPs and finds the sample complexity bounds. The paper results seem weaker than that in [6]. The paper mentions that [6] has a Lipshitz constant L. However, this constant is when the objective function and constraint function are convex, while is 1 for the setup of this paper. As such, [6] does not have the parameter L and thus the authors misrepresent the prior work. Given that the results in this work are not better than the state of the art, the reviewer does not believe that the paper can be accepted in its current form.

**Questions:**

Please see above

**Limitations:**

Please see above

**Strengths And Weaknesses:**

The paper results seem weaker than that in [6]. The paper mentions that [6] has a Lipshitz constant L. However, this constant is when the objective function and constraint function are convex, while is 1 for the setup of this paper. As such, [6] does not have the parameter L and thus the authors misrepresent the prior work. Given that the results in this work are not better than the state of the art, the reviewer does not believe that the paper can be accepted in its current form.

Based on the discussions with the authors, I am increasing the score. Some evaluations would be good in the paper. Also, more discussions on novelty would help. Use of zero constraint part achievability beyond that for relaxed setup is standard. The key contributions are: (i) lower bound on strict feasibility, (ii) achievability for relaxed setup.

---

> ### Author Response · Authors · 2022-08-02
> **Response to Reviewer KfEz - Part 1**
>
> We address the reviewer's concerns below.
>
> *The paper mentions that [6] has a Lipshitz constant L. However, this constant is when the objective function and constraint function are convex, while is 1 for the setup of this paper. As such, the AAAI version of the paper that deals with the setup as in this paper does not have the parameter L and thus the authors misrepresent the prior work.*
>
> We thank the reviewer for pointing out the AAAI version of [6] (the full version with the proofs was made available on Arxiv only on July 13). We do agree with the reviewer that $L = 1$ for the standard CMDP setting and we will correct this statement in our paper.
>
> *Given that the results in this work are not better than the state of the art, the reviewer does not believe that the paper can be accepted in its current form*
>
> In order to respond to the reviewer's claim that the results in [6] are better than ours, we do a thorough comparison and will subsequently refer to [v3] of https://arxiv.org/abs/2109.06332. [6] proposes a model-free algorithm (analogous to Wang et al, 2020) to solve the CMDP problem in both the relaxed and strict feasibility settings. Unlike standard works in the literature, [6] considers a normalized (by $1 - \gamma$) value function (defined in Eq. 1). The normalization ensures that the reward/constraint value functions lie in the $[0,1]$ range (not in the standard $[0, 1/1- \gamma] = O(H)$ range). **This difference in the scale of the value functions is the reason that the bounds in [6] seem to be better than our bounds, and we explain this in detail next.**
>
> **Relaxed feasibility setting**: In this case, [6] (Corollary 1) proves an $O(H^2 S A/\epsilon^2 \zeta^2)$ bound on the sample-complexity where $\zeta$ is the Slater-constant. When translated to the standard $O(H)$ scale of the value functions, this implies an $O(H^4 S A/\epsilon^2 \zeta^2)$ bound (see footnote 6 in [6]). In comparison, our result establishes an $O(H^3 S A/\epsilon^2)$ bound on the sample complexity (see Theorem 2). Hence, our bound is better in the dependence on $H$ (the effective horizon) and does not have a dependence on the Slater constant $\zeta$ (that can be arbitrarily close to zero). More importantly, our result implies that in the relaxed feasibility setting, solving CMDPs is as easy as solving MDPs. This is not implied by the result in [6] and is hence somewhat misleading.
>
> **Strict feasibility setting**: In this case, [6] claims that they obtain an $O(H^4 S A/\epsilon^2 \zeta^2)$ bound on the sample complexity (see Table 2 in the Appendix) for the standard $O(H)$ scaling of the value functions. This contradicts our $\Omega(H^5 S A/\epsilon^2 \zeta^2)$ lower-bound in Theorem 8. This discrepancy is again because of the scale of the constraint value functions -- in particular, if we fix the scaling in [6] by requiring the constraint values to be in the standard $[0, H]$ range, then the bound in [6] becomes $\Omega(H^6 S A/\epsilon^2 \zeta^2)$. Hence there is no violation of our lower bound and the bound in [6] is worse than our $O(H^5SA/\epsilon^2 \zeta^2)$ upper bound (see Theorem 3) by a factor of $H$. In order to further clarify this, we provide a simplified version of our lower bound.
>
> **Simplified Lower Bound**: For a target error of $\epsilon$, our lower-bound construction shows that it is important to estimate the constraint value function to a smaller error equal to $\epsilon' := \epsilon (1 - \gamma) \zeta$. This is because a small estimation error in the constraint value can (incorrectly) render the optimal policy infeasible and result in a $\frac{\text{Range(value function)}}{\zeta} = \frac{1}{(1 - \gamma)  \zeta}$ suboptimality gap. For instance, consider the 2-arm bandit setting where arm-1 has a mean reward $H$ and mean constraint reward $b-\zeta$ and arm-2 has a mean reward $0$ and mean constraint reward $b+\zeta$. For the constraint RHS equal to $b$ (implying that the Slater constant is $\zeta$), the ground truth optimal policy plays each arm with equal probability to achieve a reward value $H/2$ (with no constraint violation). However, suppose there is an error $\epsilon'$ in the estimation of the constraint reward in arm-2 (we estimate it to be $b + \zeta - \epsilon'$); then even if everything else is exactly estimated, the empirical optimal policy has to play action $2$ with probability $\zeta/(2\zeta-\epsilon')\approx 1/2 + \epsilon'/(4\zeta)$ to satisfy the constraint, which gives a value $H/2 - H\epsilon'/(4\zeta)$. This results in an $H\epsilon'/(4\zeta)$  error of the final policy. To obtain an $\epsilon$-correct policy without constraint violation, one needs to set $\epsilon'=\epsilon\zeta/H$. Combining this with the lower-bound results for unconstrained MDPs, implies that we require a sample-complexity of $\Omega(H^3 S A / \epsilon'^2) = \Omega(H^5 S A/\epsilon^2 \zeta^2)$ (this is formalized in the proof sketch of Theorem 3 (upper-bound) and Theorem 8 (lower-bound) in our paper).

---

> > ### Comment · Reviewer_KfEz · 2022-08-03
> > **Normalized vs Unnormalized setup**
> >
> > I have a few follow up questions:
> >
> > 1. If you were to get results for normalized setup, what would the results be? Would they be better than state of the art?
> >
> > 2. There seems an easy approach for unnormalized setup using normalized setup. Step 1: Normalize, Step 2: Run Algorithm for Normalized setup that achieves 0 constraint violation with \epsilon(1-\gamma) objective error. Step 3: The samples of Step 2 will give 0 constraint violation and \epsilon objective error in unnormalized setup. If we were to use that with Step 2 using the existing normalized results, the sample complexity will be lower and actually H^4. What is wrong in such an argument?

---

> > > ### Author Response · Authors · 2022-08-04
> > > **Normalized vs Unnormalized setup - Response to followup questions**
> > >
> > > 1. *If you were to get results for normalized setup, what would the results be? Would they be better than state-of-the-art?*
> > >
> > > Yes, indeed. We did mention this in our response - the last lines of the **Completing the comparison to [6]** section. If we use the normalized $[0,1]$ range for the value functions, our bound becomes $O(H S A/\epsilon^2 \zeta^2)$ which is better than the $O(H^2 S A/\epsilon^2 \zeta^2)$ bound for the same scaling in [6].
> > >
> > > 2. *What is wrong in such an argument?*
> > >
> > > There is also a factor of $\zeta_{\text{normalized}}^2$ (the Slater constant) in the denominator of the bounds. Your argument is incomplete - if we follow the above procedure we will indeed get an $O\left(\frac{H^4}{\zeta_{\text{normalized}}^2}\right)$ bound. However, the Slater constant (that measures the width of the feasible region) also depends on the range of the constraint value function and $\zeta_{\text{normalized}} = (1 - \gamma) \zeta_{\text{unnormalized}}$. Hence, after considering the scaling of the Slater constant, the total sample complexity for the above procedure is equal to $O\left(\frac{H^6}{\zeta_{\text{unnormalized}}^2}\right)$ which is worse than our $O\left(\frac{H^5}{\zeta_{\text{unnormalized}}^2}\right)$ sample-complexity bound.

---

> > > > ### Comment · Reviewer_KfEz · 2022-08-04
> > > > **Response**
> > > >
> > > > The detailed response is great. It would help the reader if some intuitions on the mentioned things in this response could be added in the revision.

---

> > > > > ### Author Response · Authors · 2022-08-04
> > > > > **Thank you for the discussion**
> > > > >
> > > > > We thank the reviewer for engaging in the discussion with us. We will definitely include the discussion and address these subtleties in the final version of the paper.

---

> ### Author Response · Authors · 2022-08-02
> **Response to Reviewer KfEz - Part 2**
>
> Continuing the reasoning from Part 1 of our response,
>
> **Completing the comparison to [6] in the strict feasibility setting**: When the value functions are normalized (like in [6]), $\text{Range(value function)} = 1$ and a small constraint violation can result in an $\frac{1}{\zeta}$ suboptimality gap. Hence, the constraint value function needs to be estimated to a smaller error equal to $\epsilon':= \epsilon \zeta$. Combining this with the standard results for the primal-dual algorithm for unconstrained MDPs (in Wang et al, 2020), this implies a sample-complexity of $O(H^2 S A / \epsilon'^2) = O(H^2 S A/\epsilon^2 \zeta^2)$ which is the sample-complexity reported in [6] (Theorem 2).
> **Translating these bounds from the normalized range to the standard $O(H)$ range for the value functions is the source of the discrepancy**. Specifically, when we use the standard $O(H)$ range for [6], $\epsilon' = \epsilon (1-\gamma) \zeta$ and using the results for the unconstrained MDPs (from Wang et al, 2020) implies an $O(H^6 S A/\epsilon^2 \zeta^2)$ sample complexity (not $O(H^4 S A/\epsilon^2 \zeta^2)$ as reported in [6]).
>
> Finally, we note that if we use the normalized $[0,1]$ range for the value functions, our bound becomes $O(H S A/\epsilon^2 \zeta^2)$ which is better than the $O(H^2 S A/\epsilon^2 \zeta^2)$ bound for the same scaling in [6].
>
> **Dependence on the number of constraints**: The bounds for both settings in [6] depend linearly on $I$ (the number of unknown constraints). Though we assume a single known constraint in our results, we can easily extend our results to handle multiple unknown constraints similar to [HasanzadeZonuzy et al, 2021]. Indeed, if the constraint rewards are unknown, a simple union bound results in a $\log(I)$ dependence on the number of constraints. This implies that our bounds will have a better dependence on the number of constraints. We will explicitly state this in the final version of the paper.
>
> **Conclusion**: To conclude, the difference between our bounds and those in [6] is because of the *scale of the value functions*. When we use the same scale (either $O(H)$ or $O(1)$) for both sets of results, our bounds are better by a factor of $H$ in both the relaxed and strict feasibility settings. Unlike [6], we provide a matching lower bound that shows an important separation in the difficulty of solving CMDPs in the relaxed vs strict feasibility settings. Furthermore, while [6] uses a specific model-free algorithm, our primal-dual algorithm involves solving a sequence of unconstrained MDPs with any black-box solver and is, therefore, more flexible. Finally, the results in [6] are only under expectation (see Theorem 2, 3), while we provide high-probability guarantees on the required sample-complexity.

---

> ### Author Response · Authors · 2022-08-09
> **Final score**
>
> Thank you again for engaging in the discussion with us and increasing your score. We notice that the contribution score is still 1 and the overall rating is 5. To recap, we have explained the difference between [6] and our paper - our upper bounds are tighter in both the relaxed and strict feasibility settings and we have matching lower bounds in both cases. Our bounds demonstrate the key difference in the hardness of solving CMDPs in the relaxed vs strict feasibility settings. Unlike [6] that directly uses the techniques in [Wang 2018], both our upper and lower bounds present significant innovations to overcome the technical difficulties. We believe that these contributions are important and worth a higher score. We kindly request the reviewer to reconsider their evaluation.

---

### Official Review · Reviewer_3jqt · 2022-07-11

**Rating:** 7
**Confidence:** 3
**Soundness:** 3 good
**Presentation:** 3 good
**Contribution:** 3 good

**Summary:**

The paper studies the sample complexity for learning constrained MDPs with a generative model. The authors provide minimax optimal upper and lower bounds for two settings: relaxed feasibility, where one seeks the best policy matching the constraints up to a slack of epsilon, and strict feasibility, where instead one focuses on strictly matching the constraints. Interestingly, the authors show that the first setting is as easy as solving unconstrained MDPs (ie the same worst-case sample complexity is attainable and optimal), while the second setting is strictly harder.

**Questions:**

Overall I think the paper provides a significant contribution to our understanding of learning constrained MDPs (though in a limited setting with a generative model). I thus vote for acceptance. Here are a few comments/questions.

1. One thing I find slightly weird is the definition of relaxed feasibility, where the same value \epsilon is used for both the objective  function and the constraint. Since r and c are different quantities with different meanings (and possibly on different scales), I was wondering whether it wouldn't be more reasonable to have two distinct parameters, say \epsilon_r and \epsilon_c, which can be used to properly tune the desired degree of sub-optimality and constraint violations.

2. Related to the previous point, if we imagine to have an algorithm and analysis for this case with two different values of \epsilon, it might be even possible to have a unified treatment for the relaxed and strict feasibility settings (right now the algorithm must be instantiated with different parameters -- eg b' -- for the two settings). In particular, the latter would be just a special case obtained by setting \epsilon_c = 0. A unified analysis might also be possible. Would there be any fundamental complication in handling two different values of epsilon for r and c?

3. Here a single constraint c is considered. What if we have multiple constraints to satisfy? How would the sample complexity scale with the number of constraints? Polynomially? Logarithmically?

4. While I find the primal-dual approach interesting, it was not completely clear to me why we cannot simply solve the empirical MDP with, say, linear programming (or any black-box approach). Could the authors elaborate on that?

5. Footnote 1 mentions that the [0,1] bounded rewards assumption is only for simplicity and one can handle other ranges. Is it also easy to handle unbounded rewards (eg sub-Gaussian)?

6. The algorithm is not really adaptive as it requires a fixed number of samples N (to collect from each s,a) as input. Would it be possible to have a variant of the algorithm with an adaptive sampling and stopping rule?

7. I guess Theorem 8 should read as "for any sound algorithm and any values of the parameters (eg S, A, etc), there exists an MDP with S states, A actions, etc such that the algorithm must pay at least the stated sample complexity". While I see how the instance in Figure 1 can be made arbitrarily large in terms of number of actions A, it is not clear there how to do the same with the number of states S.

**Limitations:**

Limitations have been properly discussed. I would stress more in the paper that considering the tabular (finite S,A) setting with a generative model limits the practical impact.

**Strengths And Weaknesses:**

Strengths:
- The paper studies a relevant problem (that of constrained MDPs)
- The results are significant as they close the gap between upper and lower bounds for this setting
- The algorithm and its analysis are, to my knowledge, novel
- The fact that CMDPs are as easy as standard MDPs in the relaxed feasibility regime is also an interesting and relevant result
- The paper is very well written. The proof sketches are very clear

Weaknesses:
- The paper assumes access to a generative model.
- The algorithm and its analysis need to handle the two settings (relaxed and strict feasibility) differently (see below).
- No numerical result is presented.

---

> ### Author Response · Authors · 2022-08-02
> **Response to Reviewer 3jqt - Part 1**
>
> We thank the reviewer for their feedback and positive review of our work. We answer their main questions below.
>
> *Since r and c are different quantities with different meanings (and possibly on different scales), I was wondering whether it wouldn't be more reasonable to have two distinct parameters, say $\epsilon_r$ and $\epsilon_c$.*
>
> Yes, that is a good suggestion. We will change the result to incorporate two distinct parameters -- $\epsilon_r$ and $\epsilon_c$ in the final version of the paper.
>
> *Iit might be even possible to have a unified treatment for the relaxed and strict feasibility settings. Would there be any fundamental complication in handling two different values of epsilon for r and c*
>
> While there is no fundamental complication in handling different values of $\epsilon_r$ and $\epsilon_c$, having a unified analysis for the two settings seems more difficult. For the relaxed feasibility case, we can use a $b'$ (RHS of the constraint in the empirical CMDP) $\leq b$. Intuitively, the (small) estimation error in $\mathcal{P}$ translates into the (small) allowed constraint violation. On the other hand, for the strict setting, since even small constraint violations are not allowed, intuitively, we need a tighter (stricter) constraint for the empirical CMDP to account for the estimation error in $\mathcal{P}$. This requires using a $b' > b$. This changes the regret decomposition for the two settings (Lemma 11 vs Lemma 12) and introduces the additional sensitivity error term for the strict feasibility setting. This additional term is responsible for the increased sample complexity in the strict feasibility setting.
>
> *Here a single constraint c is considered. What if we have multiple constraints to satisfy? How would the sample complexity scale with the number of constraints? Polynomially? Logarithmically?*
>
> Indeed, we can handle multiple unknown constraint rewards. Note that in our current bounds, we assume that the rewards and the constraint rewards are known since they do not affect the leading terms of the sample complexity (estimating $\mathcal{P}$ is the more difficult part). If the constraint rewards are unknown, similar to [HasanzadeZonuzy et al, 2021] this will result in a logarithmic dependence on the number of constraints (by a simple union bound). This is an important point, and we will make this explicit in the final version of the paper.
>
> *While I find the primal-dual approach interesting, it was not completely clear to me why we cannot simply solve the empirical MDP with, say, linear programming (or any black-box approach). Could the authors elaborate on that?*
>
> We used the primal-dual approach to obtain the optimal dependence on $H$ and $S$. It is possible to use linear programming with a model-based approach to get an $O(H^3 S^2 A/\epsilon^2)$ sample-complexity for the relaxed feasibility setting [HasanzadeZonuzy et al, 2021]. Similarly, by using a union bound over all possible value functions, it would be possible to have a black-box algorithm to get a sample complexity that scales as $O(H^3 S^2 A/\epsilon^2)$ for the relaxed feasibility setting. However, this result implies that we need to accurately estimate all values of the transition probability matrix. Since this is not necessary for the case of unconstrained MDPs (a special case of CMDPs), it is somewhat misleading.
>
> For unconstrained MDPs, getting the optimal $O(H^3 S A/\epsilon^2)$ dependence is possible by using a careful construction of an absorbing MDP (a leave-one-out analysis to avoid correlations) [Agarwal et al 2019, Li et al, 2020]. These analyses heavily rely on using dynamic programming techniques. Unfortunately, the analysis for CMDPs is more complicated because unlike unconstrained MDPs, (i) the optimal policy depends on the starting state distribution and is stochastic, and (ii) because of the interplay between the rewards and constraints, dynamic programming is not a valid approach. These difficulties prevent us from directly constructing absorbing (C)MDPs and doing a similar analysis as the unconstrained case. Our primal-dual analysis gets around this problem -- in every iteration, it requires solving an unconstrained MDP. This allows us to use the techniques from the unconstrained setting [Li et al, 2020] and obtain tight concentration bounds. Putting these results together with our regret decomposition and a union-bound over the iterations enables us to obtain the optimal sample complexity in both the relaxed and strict feasibility settings. If this intuitive explanation is helpful, we will add it to the final version of the paper.

---

> > ### Comment · Reviewer_3jqt · 2022-08-08
> > **Acknowledgement of Authors' Response**
> >
> > I thank the authors for the very detailed response. All my doubts were nicely clarified. Although, due to the reference pointed out by Reviewer KfEz, the paper is not as novel as I originally thought in terms of advancement over the state of the art, I still think this is a nice contribution and I am thus keeping my initial score.

---

> ### Author Response · Authors · 2022-08-02
> **Response to Reviewer 3jqt - Part 2**
>
> *Footnote 1 mentions that the [0,1] bounded rewards assumption is only for simplicity and one can handle other ranges. Is it also easy to handle unbounded rewards (eg sub-Gaussian)?*
>
> For our sample-complexity bounds, we assume that the reward is known (see line 113) and lies in the $[0,1]$ interval. If the $[0,1]$ range is replaced by an unbounded range, the notion of an $\epsilon$ error loses its meaning (value ranges are infinitely larger than epsilon). Any bounded range of rewards can be easily handled by rescaling. On the other hand, if the rewards are not known, but the mean rewards are in a bounded range with sub-Gaussian noise (with a common upper bound on the noise sub-gaussianity constant), then the required concentration results (and hence our bounds) can be extended to this setting.
>
> *The algorithm is not really adaptive as it requires a fixed number of samples N (to collect from each s,a) as input. Would it be possible to have a variant of the algorithm with an adaptive sampling and stopping rule?*
>
> Note that online algorithms [Ding et al, 2021; Efroni et al, 2020] used in the regret minimization setting are adaptive i.e. they adaptively query the states for a different number of samples in each iteration. Unfortunately, the resulting sample complexity for such algorithms is not optimal in the PAC setting that we consider.   On the other hand, while it should be possible to design a variant of our algorithm with adaptive sampling and an adaptive termination criterion, this would complicate the resulting analysis. We do agree that this is a good future direction of research, but it would be important to first understand how to do this in the unconstrained case. Even for unconstrained MDPs, we are not aware of a result that can adaptively query the generative model and achieve the optimal sample complexity. If the reviewer has some pointers, we would be grateful and look into this line of research.
>
> *I guess Theorem 8 should read as "for any sound algorithm and any values of the parameters (eg S, A, etc), there exists an MDP with S states, A actions, etc such that the algorithm must pay at least the stated sample complexity". While I see how the instance in Figure 1 can be made arbitrarily large in terms of number of actions A, it is not clear there how to do the same with the number of states S.*
>
> Yes, that is correct. For a more complete description of the lower-bound construction (including how to scale with the number of states), please refer to Figure 2 in Appendix E. In the final version of the paper, we will move this figure and the corresponding description to the main paper.

---

### Official Review · Reviewer_VP1s · 2022-07-11

**Rating:** 7
**Confidence:** 4
**Soundness:** 3 good
**Presentation:** 2 fair
**Contribution:** 4 excellent

**Summary:**

This work provides finite-sample analysis for constrained MDPs in a generative model/simulator setting. For the first relaxed feasibility setting, the proposed method improves the prior art sample complexity by $S$ and matches the unconstrained minimax lower bound. For a strict feasibility setting, this work provides the first lower bound and also proposed a method that can match the minimax lower bound.
Especially, this work shows that with strict feasibility requirements, CMDPs have intrinsic more challenges than unconstrained MDPs.

**Questions:**

1. Do the result need the value accuracy gap $\varepsilon$ to be the same as the small constraint violation in Line 122? From my understanding after reading the proof sketch, they can be different. But the equation (2) may lead to some confusion as they are related and must be the same. If they can be different, using different variable notation may be better.

**Limitations:**

The author already mentioned the possible extension of this work in the discussion.

**Strengths And Weaknesses:**

Advantages:
1. For the strict feasibility of CMDP settings, this work is the first one that provides a tight lower bound and also proposes a method with a minimax sample complexity bound, characterizing the difficulty of solving CMDPs theoretically.
2. Furthermore, in the relaxed feasibility setting, it also improves the prior arts and achieves the minimax lower bound (unconstrained MDPs') sample complexity.

Disadvantages:
1. The presentation of the work can be polished more by showing more intuition and key technical tools and moving the proofing details to the appendix.

---

> ### Author Response · Authors · 2022-08-02
> **Response to Reviewer VP1s**
>
> We thank the reviewer for their feedback and positive review of our work. We will improve the presentation of the paper by including more intuition for the technical results. We answer the reviewer's question below.
>
> *Do the result need the value accuracy gap to be the same as the small constraint violation in Line 122?*
>
> No, the value accuracy gap can be different from the small constraint violation in Line 122. Thank you for the suggestion. We will use a different $\epsilon_r$ and $\epsilon_c$ in the final version of the paper.

---

> ### Author Response · Authors · 2022-08-09
> **Final Score**
>
> Thank you again for a positive assessment of our work. We notice that you have decreased your overall score from 8 to 7. Could you please let us know the reason for this decrease - in particular, what additional contributions or modifications are required for a higher rating? This will help us further improve the paper. Many thanks.

---

### Meta-Review · Area_Chair_rGUf · 2022-08-22

**Recommendation:** Accept
**Confidence:** Certain

**Metareview:**

The paper studies the sample complexity of constrained MDPs and provides (nearly) matching lower and upper bounds when the constraint needs to be satisfied exactly and when slackness is allowed. Reviewer KfEz initially had concerns about comparison with prior works, which were resolved during discussion period. Other than that, multiple reviewers point out that the main text writing is not particularly informative; much space in the main text was devoted to inconsequential details while key intuitions are missing. In addition, as Reviewer 3jqt points out and the AC agrees, the generative model + tabular setting is quite restrictive and limits the significance of the work. All that said, all reviewers also agree that this is a technically solid paper with multiple interesting insights (e.g., the separation between the strict and relaxed feasibility settings), and the final reviewer scores are unanimously on the positive side.

**Award:**

No

---

### Decision · Program_Chairs · 2022-09-14

Accept